# SAV1 promotes Hippo kinase activation through antagonizing the PP2A phosphatase STRIPAK

Sung Jun Bae[1†], Lisheng Ni[1†], Adam Osinski[1], Diana R Tomchick[2], Chad A Brautigam[2,3], Xuelian Luo[1,2]*

[1]Department of Pharmacology, University of Texas Southwestern Medical Center, Dallas, United States; [2]Department of Biophysics, University of Texas Southwestern Medical Center, Dallas, United States; [3]Department of Microbiology, University of Texas Southwestern Medical Center, Dallas, United States

**Abstract** The Hippo pathway controls tissue growth and homeostasis through a central MST-LATS kinase cascade. The scaffold protein SAV1 promotes the activation of this kinase cascade, but the molecular mechanisms remain unknown. Here, we discover SAV1-mediated inhibition of the PP2A complex STRIPAK[SLMAP] as a key mechanism of MST1/2 activation. SLMAP binding to autophosphorylated MST2 linker recruits STRIPAK and promotes PP2A-mediated dephosphorylation of MST2 at the activation loop. Our structural and biochemical studies reveal that SAV1 and MST2 heterodimerize through their SARAH domains. Two SAV1–MST2 heterodimers further dimerize through SAV1 WW domains to form a heterotetramer, in which MST2 undergoes trans-autophosphorylation. SAV1 directly binds to STRIPAK and inhibits its phosphatase activity, protecting MST2 activation-loop phosphorylation. Genetic ablation of SLMAP in human cells leads to spontaneous activation of the Hippo pathway and alleviates the need for SAV1 in Hippo signaling. Thus, SAV1 promotes Hippo activation through counteracting the STRIPAK[SLMAP] PP2A phosphatase complex.

DOI: https://doi.org/10.7554/eLife.30278.001

*For correspondence:
xuelian.luo@utsouthwestern.edu

†These authors contributed equally to this work

Competing interests: The authors declare that no competing interests exist.

## Introduction

The balance between cell division and death maintains tissue homeostasis of multicellular organisms. Through restricting cell proliferation and promoting apoptosis, the evolutionarily conserved Hippo pathway is critical for tissue homeostasis and tumor suppression (*Harvey and Tapon, 2007*; *Johnson and Halder, 2014*; *Pan, 2010*; *Yu et al., 2015*). The role of the Hippo pathway in tumorigenesis is well-established in model organisms, such as fly and mouse (*Halder and Johnson, 2011*; *Pan, 2010*). Aberrant expression and genetic mutations of Hippo pathway components have been linked to human cancers (*Pan, 2010*; *Yu et al., 2015*). A better molecular understanding of the Hippo pathway may uncover new targets for cancer therapy.

In mammals, the canonical Hippo pathway contains a core kinase cascade, formed by STE20-like MST1/2 kinases, Large Tumor Suppressor LATS1/2 kinases, the WW-domain containing scaffold protein Salvador (SAV1) and the adaptor protein MOB1 (*Halder and Johnson, 2011*; *Pan, 2010*; *Yu and Guan, 2013*). Upon activation by extracellular signals, such as cell-cell contact and mechanical cues, cell-surface receptors transduce signals to a cytoskeletal complex containing the FERM proteins NF2 and Expanded (EX), which relays these upstream signals to activate MST1/2 (*Hamaratoglu et al., 2006*; *Zhang et al., 2010*). Activated MST1/2 partner with SAV1 to phosphorylate and activate the LATS1/2–MOB1 complexes (*Chan et al., 2005*; *Praskova et al., 2008*), which in turn phosphorylate and inactivate the oncogenic transcriptional co-activators YAP/TAZ

(*Dong et al., 2007*; *Liu et al., 2010*; *Zhao et al., 2010*; *Zhao et al., 2007*). LATS1/2–MOB1-mediated phosphorylation inhibits YAP/TAZ by promoting their nuclear export, cytoplasmic sequestration, and ubiquitination-dependent degradation. When Hippo signaling is turned off, YAP/TAZ translocate to the nucleus and bind to the TEAD family of transcription factors to promote the transcription of pro-proliferative and pro-survival genes, enabling cell proliferation (*Luo, 2010*; *Zhao et al., 2008b*).

Although the MAP4K family kinases can act as additional activators of LATS1/2 in response to certain upstream signals (*Li et al., 2014*; *Meng et al., 2015*; *Plouffe et al., 2016*; *Zheng et al., 2015*), genetic studies in fly and mouse have established the requirement for the canonical MST-LATS kinase cascade in tumor suppression (*Harvey et al., 2003*; *Justice et al., 1995*; *Kango-Singh et al., 2002*; *Lai et al., 2005*; *Tapon et al., 2002*; *Wu et al., 2003*; *Xu et al., 1995*). Activation of MST1/2 is sufficient to turn on the Hippo pathway to inhibit YAP/TAZ-mediated gene expression (*Song et al., 2010*; *Zhou et al., 2009*). It is critical to understand the mechanism of MST1/2 kinase activation, a key initiating event in Hippo signaling. MST1/2 contain an N-terminal kinase domain and a C-terminal coiled-coil SARAH domain, which are separated by a flexible linker. MST1 and MST2 can each form a constitutive homodimer through the SARAH domain, and kinase activation is achieved by trans-autophosphorylation of the activation loop (T183 for MST1 and T180 for MST2) (*Avruch et al., 2012*; *Jin et al., 2012*; *Ni et al., 2013*). Although MST1/2 form homodimers and undergo constitutive autophosphorylation in vitro, the vast majority of MST1/2 molecules remain unphosphorylated at the T-loop in vivo when the Hippo pathway is off (*Praskova et al., 2004*), suggesting that MST1/2 activation is antagonized by phosphatases. Recent studies have suggested that Hippo activity is negatively regulated by a multi-subunit protein phosphatase 2A (PP2A) complex, called striatin-interacting phosphatase and kinase (STRIPAK) in *Drosophila* (*Ribeiro et al., 2010*). Furthermore, MST1/2 interact with STRIPAK through the adaptor protein SLMAP in human cells (*Couzens et al., 2013*; *Hauri et al., 2013*), although the mechanism and functional consequence of this interaction are unclear.

Among the core components of the MST-LATS cascade, MOB1 acts as a dynamic phospho-peptide-binding adaptor to promote MST1/2-dependent activation of LATS1/2 (*Ni et al., 2015*). By contrast, the mechanism by which the other core component SAV1 promotes MST1/2 activation remains unclear (*Pantalacci et al., 2003*; *Udan et al., 2003*; *Wu et al., 2003*). SAV1 contains an N-terminal flexible region that binds to the FERM domain of NF2 (*Yu et al., 2010*), two WW domains, and a C-terminal SARAH domain. Binding of SAV1 to MST1/2 is mediated by their SARAH domains (*Callus et al., 2006*). One attractive hypothesis is that the SARAH domains of SAV1 and MST1/2 form a heterotetramer to strengthen the MST homodimer, thus promoting MST1/2 trans-autophosphorylation at the T-loop (*Ni et al., 2013*). It has also been suggested that SAV1 recruits MST1/2 to the plasma membrane for activation (*Harvey and Tapon, 2007*; *Yin et al., 2013*).

In this study, using a combination of biochemical, structural, and cell biological experiments, we establish the PP2A complex, STRIPAK^SLMAP, as a key negative regulator of Hippo signaling in human cells, and show that SAV1 promotes MST1/2 activation through antagonizing the STRIPAK^SLMAP PP2A phosphatase activity.

## Results

### Feedback inhibition of MST2 activation by autophosphorylated MST2 linker

MST2 undergoes autophosphorylation at multiple threonine-methionine (TM) motifs in its linker (*Figure 1A*) (*Ni et al., 2015*). One such motif, pT378M, acts as the primary docking site for MOB1. Binding of MST2 converts MOB1 to the open conformation, allowing MOB1 to bind and recruit LATS1 to MST2 for phosphorylation. Therefore, phosphorylation of the MST2 linker is critical for activation of the Hippo pathway. Surprisingly, we found that mutation of 7 TMs to AMs (7TA) in MST2 not only abolished MOB1 binding to MST2, but also dramatically increased MST2 T180 phosphorylation in human cells (*Figure 1B*), suggesting that linker phosphorylation could inhibit MST1/2 kinase activation in a feedback mechanism. This result is consistent with earlier findings that truncation of the MST1/2 linker increases MST1/2 kinase activity (*Creasy et al., 1996*). Therefore,

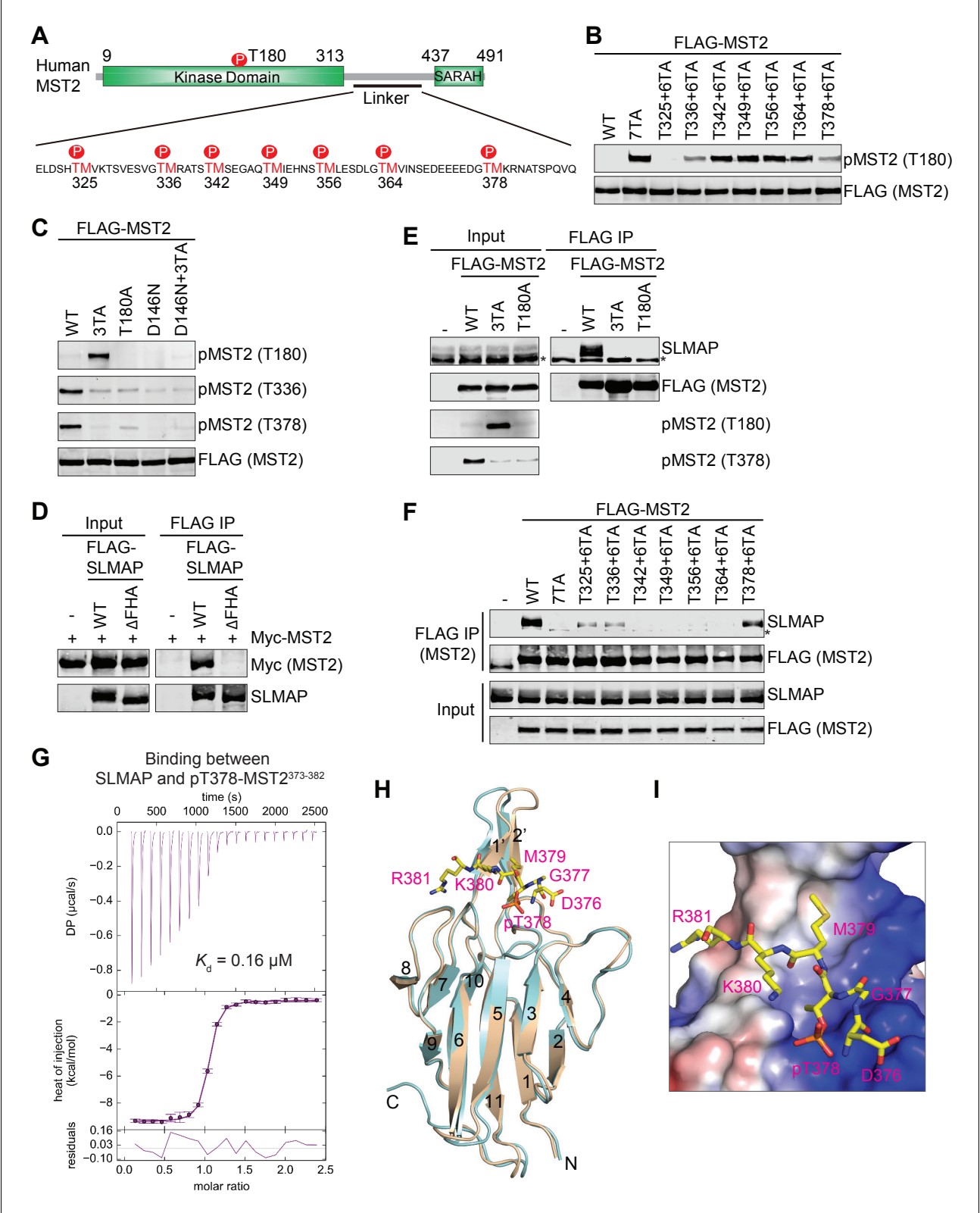

**Figure 1.** Feedback inhibition of MST2 activation by SLMAP binding to autophosphorylated MST2 linker. (**A**) Domain organization of human MST2 and seven phospho-TM sites in the linker region. (**B**) Identification of the TM motifs in the MST2 linker critical for inhibition of MST2 T180 phosphorylation. Anti-FLAG and anti-pMST2 (T180) blots of lysates of 293FT cells transfected with the indicated FLAG-MST2 plasmids. Anti-FLAG blot was used to evaluate protein expression levels. (**C**) Anti-FLAG, anti-pMST2 (T180), anti-pMST2 (T336), and anti-pMST2 (T378) blots of lysates of 293FT cells

*Figure 1 continued on next page*

*Figure 1 continued*

transfected with the indicated FLAG-MST2 plasmids. (**D**) Anti-Myc and anti-SLMAP blots of cell lysates (input) and anti-FLAG immunoprecipitates (IP) of 293FT cells transfected with the indicated plasmids. WT, wild type; ΔFHA, mutant with FHA deleted. (**E**) Immunoblots of cell lysates (input) and anti-FLAG IP of 293FT cells transfected with the indicated plasmids. Endogenous SLAMP was detected by the anti-SLMAP blot. Asterisk designates a non-specific band. (**F**) Immunoblots of cell lysates (input) and anti-FLAG IP of 293FT cells transfected with the indicated plasmids. Asterisk designates a non-specific band. (**G**) ITC thermogram (top) and isotherm (middle) of the binding between purified recombinant human SLMAP FHA and the pT378-MST2$^{373-382}$ (pMST2) peptide with the binding affinity ($K_d$) indicated. DP, differential power. (**H**) Superposition of the crystal structures of human SLMAP FHA (cyan) and SLMAP FHA in complex with the pMST2 peptide (wheat). pMST2 is colored yellow and shown as sticks. The pMST2 residues are labeled in magenta. All structure figures are generated by PyMOL (Schrödinger, LLC; http://www.pymol.org). (**I**) Surface drawing of the pT-binding site of SLMAP FHA. The surface is colored according to the electrostatic potential, with blue, red, and white representing positive, negative, and neutral charges, respectively. pMST2 is shown as yellow sticks with residues labeled.

DOI: https://doi.org/10.7554/eLife.30278.002

The following figure supplements are available for figure 1:

**Figure supplement 1.** Feedback inhibition of MST2 activation by SLMAP binding to autophosphorylated MST2 linker.
DOI: https://doi.org/10.7554/eLife.30278.003

**Figure supplement 2.** Crystal structures of SLMAP FHA and SLMAP FHA bound to pMST2.
DOI: https://doi.org/10.7554/eLife.30278.004

autophosphorylation of the MST2 linker has dual, opposing functions in Hippo signaling: recruitment and activation of MOB1 for downstream signaling and feedback inhibition of MST2 activation.

To identify which TM motif mediated feedback inhibition of MST2 activation, we mutated each TM to AM, and found that none of the single TA mutant had highly elevated T180 phosphorylation, suggesting that more than one pTM site were involved in MST2 inactivation in human cells (*Figure 1—figure supplement 1A*). Restoration of T325, T336, and T378 individually to the 7TA mutant reduced T180 phosphorylation to various levels, indicating that phosphorylation of these three TM sites is critical for MST2 inactivation (*Figure 1B*). The MST2 3TA mutation with these three threonine residues mutated to alanine was as efficient as 7TA in activating MST2 (*Figure 1—figure supplement 1B*). We previously raised a phospho-specific antibody against the pT378 site (*Ni et al., 2015*). To confirm that T336 was phosphorylated in full-length MST2, we then raised a phospho-specific antibody against the pT336 site. We showed that both T336 and T378 sites were indeed phosphorylated in wild type (WT) MST2, but not in 3TA (*Figure 1C* and *Figure 1—figure supplement 1C*). Moreover, phosphorylation at T336 and T378 and the enhanced phosphorylation of T180 in 3TA were abolished in the kinase-inactive MST2 mutants (T180A and D146N) in human cells (*Figure 1C*). Active MST2 kinase domain efficiently phosphorylated T180, T336, and T378 of kinase-dead MST2 D146N in vitro (*Figure 1—figure supplement 1D*). These results indicate that the inhibitory phosphorylation at the MST2 linker occurs through autophosphorylation.

## SLMAP binds to phosphorylated MST2 linker

Because deletion of the MST2 linker does not enhance either the activation or kinase activity of recombinant purified MST2 in vitro (*Figure 1—figure supplement 1E*) (*Ni et al., 2013*), we hypothesized that the inhibitory pTM motifs recruited a phosphatase to dephosphorylate pT180 and inactivate MST2. A proteomic study has shown that MST1/2 interact with a specific PP2A complex called STRIPAK through the N-terminal forkhead associated (FHA) domain of the adaptor protein SLMAP (*Couzens et al., 2013*). Consistent with this previous report, we showed that MST2 interacted with SLMAP WT, but not SLMAP ΔFHA, in human cells (*Figure 1D*). SLMAP interacted with MST2 WT, but not 3TA and T180A mutants (*Figure 1E*). Moreover, SLMAP bound to MST2 mutants with a single pTM site restored at pT325, pT336, or pT378 to varying degrees in human cells (*Figure 1F*). Thus, the inhibitory phosphorylation at these three sites in the MST2 linker is required for the SLMAP-MST2 interaction. Recombinant GST-SLMAP FHA interacted with the full-length (FL) MST2, but not the MST2 kinase domain (KD) or MST2 with the linker deleted (ΔL), suggesting that SLMAP FHA binds directly to the MST2 linker in vitro (*Figure 1—figure supplement 1F*). As measured by isothermal titration calorimetry (ITC), the binding affinities between SLMAP FHA and phospho-MST2 peptides containing pT325, pT336, or pT378 were 0.39 μM, 1.1 μM, and 0.16 μM, respectively (*Figure 1G* and *Table 1*). Collectively, our data suggest that SLMAP binding to the phospho-MST2 linker recruits STRIPAK to MST2 and promotes PP2A-mediated dephosphorylation of pT180.

**Table 1.** Summary of the ITC results.

| Protein | pMST2 peptide | $K_d$ (μM) | ΔH (kcal/mol) | ΔS (cal/mol·K) |
|---|---|---|---|---|
| MOB1[33-216] | pT378-MST2[373-382] | 9.95 | −4.0 | 9.2 |
| | pT378-MST2[371-401] | 0.296 | −13.5 | −16.1 |
| SLMAP FHA | pT325-MST2[320-329] | 0.392 | −10.3 | −5.8 |
| | pT336-MST2[331-340] | 1.14 | −8.2 | −0.7 |
| | pT378-MST2[373-382] | 0.160 | −9.0 | 0.5 |
| | pT378-MST2[371-401] | 0.393 | −9.5 | −3.2 |
| SLMAP FHA/R32A | pT378-MST2[373-382] | 18.8 | −6.7 | −1.3 |

DOI: https://doi.org/10.7554/eLife.30278.005

Interestingly, MOB1 and SLMAP bind to the pT378M site (the pT378-MST2[371-401] peptide) with similar affinities (*Table 1*), suggesting that MOB1 may contribute to MST1/2 activation through competing with SLMAP for MST1/2 binding.

## Structural basis of phospho-MST2 binding by SLMAP

FHA domains are specific phospho-threonine (pT) binding modules (*Mahajan et al., 2008*). We obtained crystals of free human SLMAP FHA domain that diffracted to 1.08 Å resolution and determined the structure with molecular replacement using the FHA of the centrosomal protein CEP170 as the search model (PDB ID: 4JON) (*Figure 1—figure supplement 2A* and *Table 2*). Like other FHA domains, SLMAP FHA adopts a typical β sandwich fold that consists of two large β sheets. One sheet contains six antiparallel strands: β2, β1, β11, β10, β7, and β8, while the other sheet contains five mixed parallel/antiparallel strands: β4, β3, β5, β6, and β9. The major structural differences between SLMAP FHA and other FHAs with known structures reside in the loops connecting the β strands, especially the long loop between β10 and β11, which forms a short β1'/β2' hairpin and protrudes away from the FHA core.

To understand how SLAMP FHA recognized phospho-MST2, we obtained crystals of SLMAP FHA bound to the pT378-MST2[371-401] peptide (pMST2 hereafter for simplicity) that diffracted to 1.55 Å resolution. We determined the structure of SLMAP FHA–pMST2 by molecular replacement using apo-SLMAP FHA as the search model (*Figure 1—figure supplement 2B* and *Table 2*). In the SLMAP FHA–pMST2 structure, the FHA domain adopts a conformation virtually identical to that of free SLMAP FHA with an RMSD of 0.33 Å for all atoms (*Figure 1H*), indicating that binding of pMST2 does not appreciably alter the conformation of the FHA domain. Only six residues of pMST2 (D[376]GpTMKR[381]) have clear electron density, and bind to a surface formed by residues from the β3-β4, β4-β5, β6-β7, and β10-β11 loops of SLMAP FHA (*Figure 1H* and *Figure 1—figure supplement 2C*). The pT378 residue binds at a highly positively charged pocket, with the phosphate group making favorable electrostatic interactions with side chain or main chain atoms from R32, S52, R53 and S75 of SLMAP (*Figure 1I* and *Figure 1—figure supplement 2C*). M379 inserts into an adjacent hydrophobic pocket on SLMAP. R381 makes hydrophobic and electrostatic interactions with side chains from V107 and V109, and main chain from D108 of SLMAP, respectively, indicating that a positively charged residue is preferred at this position. D376 forms two hydrogen bonds with R53 of SLMAP. G377 forms a 90° turn to allow D376 to make favorite contacts with R53 of SLMAP. Our structure thus suggests that SLMAP FHA prefers binding to a phospho-threonine peptide with a consensus sequence of E/D-G-pT-M-x-R/K. Consistent with this prediction, the pT325-MST2[320-329] and pT336-MST2[331-340] peptides that do not fully conform to this consensus bind to SLMAP FHA with lower affinities (*Table 1*).

Despite the differences of the loops at the pT binding site among various FHA domains, the pT-binding residues, R32, S52 and H55, can be structurally overlaid with the corresponding residues from other FHAs, suggesting that the mode of pT recognition is highly conserved among FHA domains. Indeed, the binding affinity between SLMAP FHA/R32A mutant and pMST2 was reduced by ~120 fold (*Table 1*). Mutations of these three conserved residues in the context of full-length SLMAP disrupted binding to MST2 in human cells (*Figure 1—figure supplement 2D*), validating the importance of the pT-binding site of SLMAP in MST2 binding.

**Table 2.** Data collection and refinement statistics for apo-SLMAP FHA and the SLMAP FHA–pMST2 complex.

**Data collection**

| Crystal | Apo | Complex |
|---|---|---|
| Space group | $P2_12_12_1$ | $P2_12_12_1$ |
| Wavelength (Å) | 0.97918 | 0.97918 |
| Unit cell | | |
| $a$, $b$, $c$ (Å) | 42.90, 51.22, 56.42 | 38.76, 70.53, 91.02 |
| Resolution range (Å) | 50–1.08 (1.10–1.08)* | 38.24–1.55 (1.59–1.55) |
| Unique reflections | 53,830 (2,647) | 36,933 (2,719) |
| Multiplicity | 8.6 (4.5) | 12.5 (10.3) |
| Data completeness (%) | 99.7 (98.5) | 99.8 (97.3) |
| $R_{merge}$ (%)[†] | 5.8 (53.9) | 10.2 (107.2) |
| $R_{pim}$ (%)[‡] | 2.0 (23.6) | 2.9 (31.8) |
| $I/\sigma(I)$ | 43.9 (2.3) | 39.4 (1.4) |
| $CC_{1/2}$[§] | 0.847 | 0.784 |
| Wilson $B$-value (Å$^2$) | 9.92 | 22.70 |
| Refinement statistics | | |
| Resolution range (Å) | 20.49–1.08 (1.11–1.08) | 38.24–1.55 (1.59–1.55) |
| No. of reflections $R_{work}/R_{free}$ | 53,761/2,000 (3,571/139) | 35,078/1,846 (2,583/136) |
| Data completeness (%) | 99.6 (97.0) | 99.7 (97.3) |
| Atoms (non-H protein/solvent/peptide) | 1,325/235/0 | 2,210/203/102 |
| $R_{work}$ (%) | 16.4 (26.6) | 16.1 (27.5) |
| $R_{free}$ (%) | 17.9 (27.7) | 18.8 (30.3) |
| R.m.s.d. bond length (Å) | 0.007 | 0.009 |
| R.m.s.d. bond angle (°) | 0.935 | 1.005 |
| Mean $B$-value (Å$^2$) (protein/solvent/peptide) | 12.83/25.96/- | 35.93/39.37/48.18 |
| Ramachandran plot (%) (favored/additional/disallowed)[#] | 97.1/2.9/0.0 | 97.4/2.6/0.0 |

*Data for the highest-resolution shell are shown in parentheses.

[†]$R_{merge} = 100\ \Sigma_h\Sigma_i|I_{h,i} - \langle I_h\rangle|/\Sigma_h\Sigma_i\langle I_{h,i}\rangle$, where the outer sum (h) is over the unique reflections and the inner sum (i) is over the set of independent observations of each unique reflection.

[‡]$R_{pim} = 100\ \Sigma_h\Sigma_i\ [1/(n_h - 1)]^{1/2}|I_{h,i} - \langle I_h\rangle|/\Sigma_h\Sigma_i\langle I_{h,i}\rangle$, where $n_h$ is the number of observations of reflections h.

[§]$CC_{1/2}$ values shown are for the highest resolution shell.

[#]As defined by the validation suite MolProbity.

DOI: https://doi.org/10.7554/eLife.30278.006

## STRIPAK[SLMAP] inhibits Hippo signaling in human cells

To determine the functional significance of the MST–SLMAP interaction in vivo, we depleted SLMAP in 293FT cells by RNA interference (RNAi). SLMAP depletion increased pT180 of MST2 WT (*Figure 2A*). SLMAP-binding-deficient MST2 3TA mutant had a pT180 level similar to that of MST2 WT in SLMAP RNAi cells. Depletion of SLMAP did not further increase pT180 of MST2 3TA. Depletion of STRIP1, a core component of the STRIPAK complex (*Goudreault et al., 2009*), similarly increased T180 phosphorylation of MST2, and this increase was blocked by the ectopic expression of RNAi-resistant FLAG-STRIP1 (*Figure 2—figure supplement 1A*). These results suggest that STRIPAK[SLMAP] is involved in suppressing MST2 activation.

We next deleted *SLMAP* from 293FT and MCF10A cells with the CRISPR (Clustered regularly interspaced short palindromic repeats)/Cas9 system. Compared to control cells, SLMAP knockout (KO) cells at low densities showed increased T-loop phosphorylation of MST1/2 and elevated MOB1 phosphorylation at T35 (*Figure 2B and C*). Moreover, phosphorylation of LATS1/2 hydrophobic motif (HM) and YAP was also increased in these cells without contact inhibition. Consistent with the

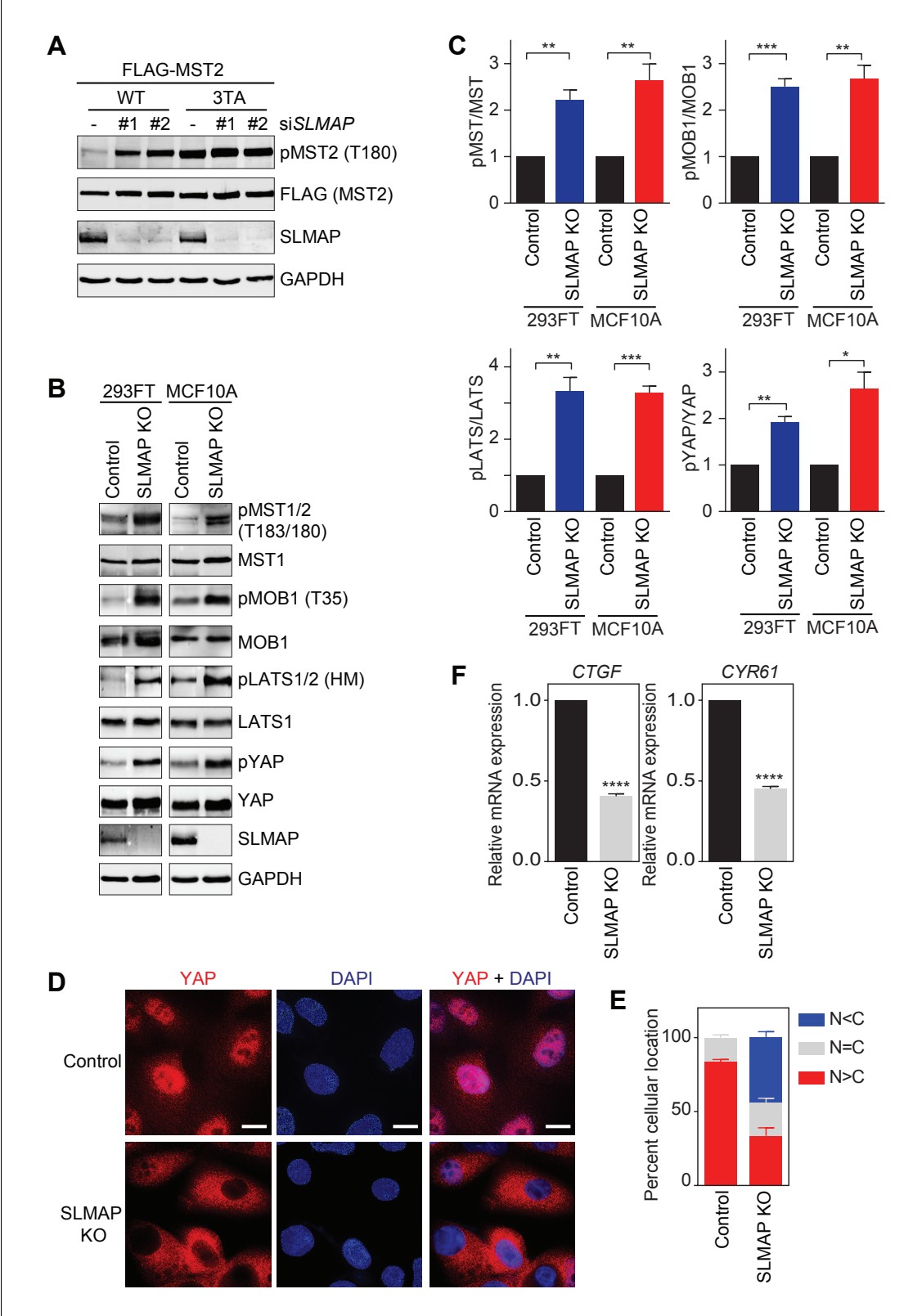

**Figure 2.** STRIPAK[SLMAP] inhibits the Hippo pathway in human cells. (**A**) 293FT cells were co-transfected with siSLMAP and the indicated FLAG-MST2 plasmids. The total cell lysates were blotted with the indicated antibodies. Anti-GAPDH blot was used as the loading control. (**B**) Immunoblots with the indicated antibodies of lysates of 293FT and MCF10A cells with SLMAP deleted. KO, knockout; HM, hydrophobic motif. (**C**) Quantification of the ratios of pMST/MST, pMOB1/MOB1, pLATS/LATS, and pYAP/YAP signals in (**B**). The total and phosphorylated protein levels were individually normalized to
*Figure 2 continued on next page*

*Figure 2 continued*

GAPDH levels. Normalized values were used to calculate the ratios. Data are plotted as mean ± SEM of three biological replicates (*p<0.05; **p<0.01; ***p<0.001). (**D**) Immunofluorescence staining of YAP localization in control and SLMAP KO MCF10A cells. Cells were fixed, permeabilized, and stained with anti-YAP (red) and DAPI (blue). Scale bars, 10 μm. (**E**) Quantification of immunofluorescence signal intensities in (**D**). Approximately 50 cells were counted from 7 random fields. N < C (blue), N = C (grey), and N > C (red) categories indicate YAP localization in cytoplasm, both cytoplasm and nucleus, and nucleus, respectively. Data are plotted as mean ± SEM of three biological replicates. (**F**) Relative expression of YAP target genes *CTGF* and *CYR61* in control and SLMAP KO MCF10A cells. Data are plotted as mean ± SEM of three biological replicates (****p<0.0001).

DOI: https://doi.org/10.7554/eLife.30278.007

The following figure supplement is available for figure 2:

**Figure supplement 1.** STRIPAK[SLMAP] inhibits the Hippo pathway in human cells.

DOI: https://doi.org/10.7554/eLife.30278.008

spontaneous activation of the Hippo pathway, a higher percentage of SLMAP KO cells exhibited cytoplasmic localization of YAP (*Figure 2D and E*). As determined by real-time quantitative PCR (qPCR), expression of two well-established Hippo target genes, Connective Tissue Growth Factor (*CTGF*) and Cysteine Rich Angiogenic Inducer 61 (*CYR61*) (*Zhao et al., 2008a*), was reduced in SLMAP KO cells (*Figure 2F*). Furthermore, expression of SLMAP WT, but not ΔFHA, rescued the phenotypes of SLMAP KO cells (*Figure 2—figure supplement 1B,C*). These results establish STRIPAK[SLMAP] as a key negative regulator of Hippo signaling in human cells. Inactivation of STRIPAK[SLMAP] leads to spontaneous activation of the Hippo pathway without contact inhibition or other upstream signals.

## SAV1 is required for Hippo pathway activation in human cells

To validate the role of SAV1 in the Hippo pathway, we generated SAV1 knockout (KO) MCF10A cells with CRISPR/Cas9. Latrunculin B (LatB) treatment is a well-established condition that activates the Hippo pathway through actin depolymerization. However, the involvement of MST1/2 in regulation of LATS1/2 and YAP by cytoskeleton reorganization has been suggested to be dependent on cell types (*Zhao et al., 2012*). We show that addition of LatB to MCF10A cells induced MST1/2 T-loop phosphorylation and MOB1 T35 phosphorylation, as well as downstream phosphorylation of LATS1/2 and YAP (*Figure 3A*), indicating that MST1/2 activation can turn on Hippo signaling through disruption of the actin cytoskeleton in human cells. In addition, the levels of all these phosphorylation events were reduced in SAV1 KO cells with or without LatB treatment (*Figure 3B*). Consistent with the defective MST-LATS kinase activation, the relative mRNA levels of Hippo target genes, *CTGF* and *CYR61*, in SAV1 KO cells were significantly higher than those in control cells, with or without LatB treatment (*Figure 3C*). These data indicate that SAV1 is required for efficient MST1/2 activation during Hippo signaling in human cells.

SAV1 and MST1/2 interact through their SARAH domains. We have shown previously that the recombinant MST2-SAV1 complex forms a heterotetramer in vitro (*Ni et al., 2013*). One possibility is that two SAV1 SARAH domains bind to one MST2 SARAH homodimer to further stabilize the MST2 homodimer, thereby enhancing the trans-autophosphorylation at the T-loop of MST2 and increasing its kinase activity. Unexpectedly, a recombinant SAV1 fragment containing the SARAH domain did not stimulate the MST2 kinase activity toward MOB1 in vitro (*Figure 3D*). Thus, SAV1 binding to MST2 does not directly stimulate its kinase activity. Instead, SAV1 promotes MST1/2 activation in human cells likely through an indirect mechanism.

## Crystal structure of the MST2–SAV1 complex

SAV1 contains an N-terminal flexible region, two WW domains, and a SARAH domain (*Figure 4A*). To understand how SAV1 activates MST2, we determined the crystal structure of MST2 ΔL/D146N bound to the SARAH domain of SAV1 (residues 291–383) (*Figure 4B* and *Table 3*). Surprisingly, MST2–SAV1 does not form an expected heterotetramer through a four-helix bundle formed by their SARAH domains. Instead, SAV1 SARAH and MST2 SARAH form a heterodimer, similar to that formed by the SARAH domains of MST2 and one of the RASSF family of tumor suppressors, RASSF5 (*Figure 4C and D*) (*Ni et al., 2013*). The active site of MST2 in MST2–SAV1 is well-defined. The $Mg^{2+}$ ion and AMP-PNP are clearly visible in the electron-density map. The MST2 kinase domain in MST2–SAV1 is structurally very similar to the MST2 kinase alone, except that the T-loop helix

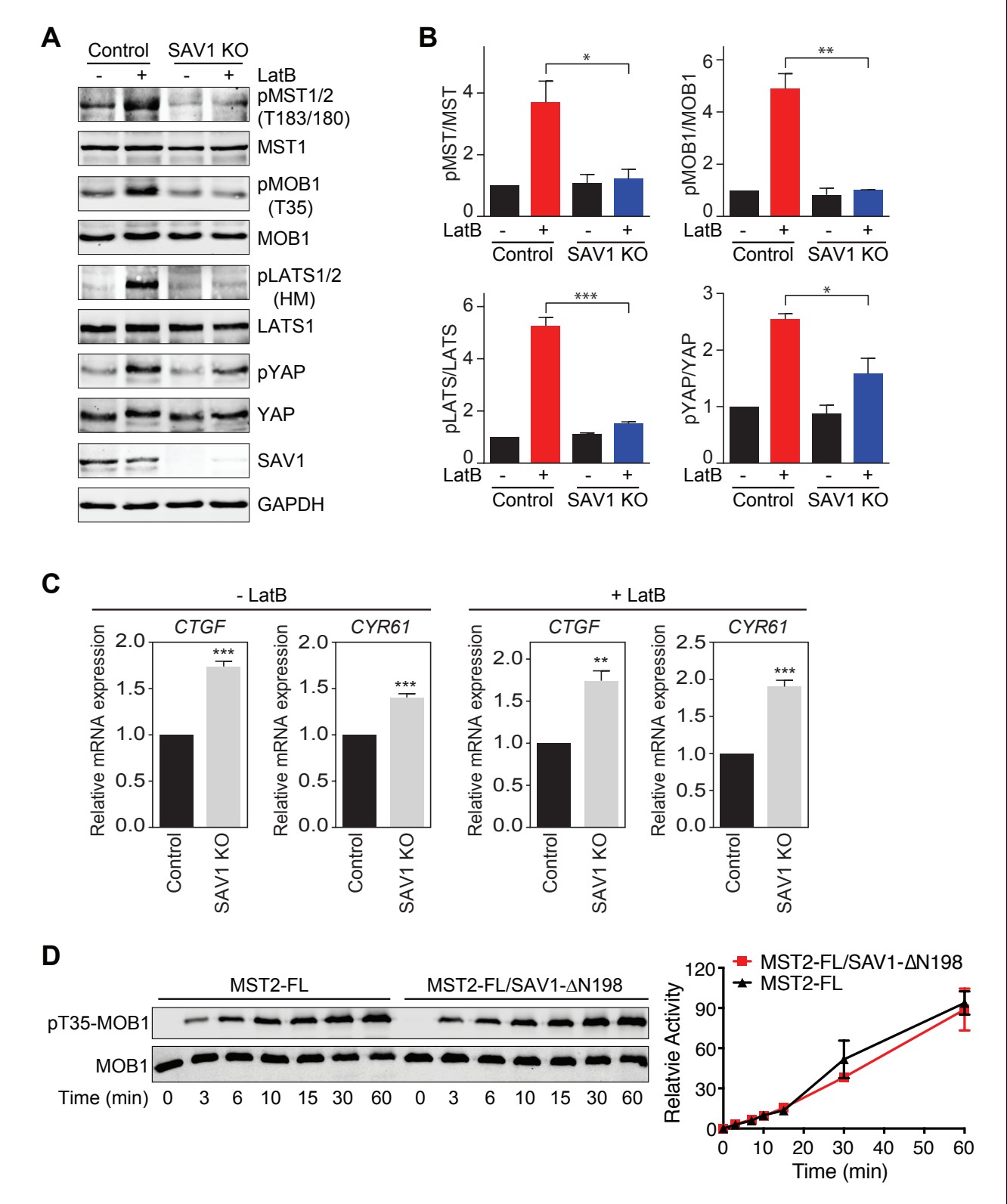

**Figure 3.** SAV1 is required for Hippo pathway activation in human cells. (**A**) Immunoblots of cell lysates of MCF10A and MCF10A-SAV1 KO cells with or without latrunculin B (LatB) treatment with the indicated antibodies. Both control and SAV1 KO cells were treated with LatB (1 µg/ml) or vehicle for 1 hr before they were harvested. (**B**) Quantification of the ratios of pMST/MST, pMOB1/MOB1, pLATS/LATS, and pYAP/YAP signals in (**A**). The total and phosphorylated protein levels were individually normalized to GAPDH levels. Normalized values were used to calculate the ratios. Data are plotted as

*Figure 3 continued on next page*

Figure 3 continued

mean ± SEM of three biological replicates (*p<0.05; **p<0.01; ***p<0.001). (C) Relative expression of YAP target genes *CTGF* and *CYR61* in control and SAV1 KO MCF10A cells with or without LatB treatment. Data are plotted as mean ± SEM of three biological replicates (**p<0.01 and ***p<0.001). (D) Quantitative immunoblots with the indicated antibodies of the in vitro kinase reactions containing human MOB1 and the indicated MST2-FL or MST2-FL/SAV1-ΔN198 proteins at the indicated times. The relative pT35-MOB1 signal intensities, normalized to those of MST2 FL and MST2-FL/SAV1-ΔN198 at 60 min (100%), respectively, are plotted against time. Means ± range for two biological replicates are plotted.
DOI: https://doi.org/10.7554/eLife.30278.009

observed in the MST2 kinase structure is not visible in the MST2–SAV1 structure. The K56-E70 salt bridge between strand β3 and helix αC is not formed in MST2–SAV1, consistent with MST2 adopting the inactive conformation. The SARAH domains of MST2 and SAV1 form a long antiparallel coiled coil and do not directly contact the kinase domain. The N-terminal extension of SAV1 SARAH forms two short helices, H0 and H1, which interact with the main coiled coil through several hydrophobic interactions. The *Drosophila* RASSF ortholog (dRASSF) has previously been shown to restrict Hippo (The *Drosophila* MST1/2 ortholog) activity by competing with SAV for binding to Hippo, and dRASSF-associated Hippo is inactive whereas SAV-associated Hippo is active (*Polesello et al., 2006*). Consistently, RASSF5 and RASSF1 (the founding member of the RASSF family) can suppress MST1 activation both in vitro and in mammalian cells (*Praskova et al., 2004*). Therefore, RASSF and SAV regulate Hippo/MST activation by forming different SARAH-domain-dependent complexes. Interestingly, the MST2-binding interface residues are conserved between SAV1 SARAH and RASSF5 SARAH (*Figure 4E*). Furthermore, the SAV1- and RASSF5-binding surfaces of MST2 SARAH are virtually identical (*Figure 4C and D*), readily explaining mutually exclusive binding of SAV1 and RASSF5 to MST2. Thus, our data support the notion that RASSFs antagonize SAV function in Hippo/MST activation.

To validate the interface between the MST2 and SAV1 SARAH domains observed in our structure, we systematically mutated the interface residues in the SARAH domain of SAV1 and tested the effects of these mutations in an in vitro binding assay (*Figure 4F* and *Figure 4—figure supplement 1A*). GST-MST2 SARAH efficiently pulled down [35]S-labeled Myc-SAV1 SARAH obtained through in vitro translation. Among the 14 mutants, SAV1 E346A, I350A, Y357A, R358A, L361A, L365A, and R368A completely lost the binding to MST2. Several other mutants also exhibited weaker binding to MST2. Thus, the MST2–SAV1 interface revealed by our crystal structure is critical for mediating the MST2–SAV1 interaction.

## SAV1 WW domains mediate the formation of the MST2-SAV1 heterotetramer

The SARAH domains of MST2 and SAV1 can only form a heterodimer, but not a heterotetramer. Yet, MST2 and larger fragments of SAV1 can form a heterotetramer based on gel filtration chromatography, suggesting that other regions of SAV1 could self-associate. Consistent with this notion, a deletion mutant of SAV1 without the intact SARAH domain has been reported to form a homodimer (*Callus et al., 2006*). Both analytical ultracentrifugation (AUC) and gel-filtration experiments confirmed that a SAV1 fragment containing both WW domains and the SARAH domain associated with MST2 in a complex consistent with a heterotetramer (*Figure 4G* and *Figure 4—figure supplement 1B*). A shorter fragment of SAV1 lacking the WW domains formed a simple heterodimer with MST2. Thus, SAV1 WW domains mediate heterotetramerization of MST2–SAV1 (*Figure 4H*).

WW2 of mouse SAV1 has been shown to form a homodimer in solution (*Figure 4—figure supplement 1C*) (*Ohnishi et al., 2007*). Human SAV1 WW2 shares 97% sequence identity with mouse SAV1 WW2, and can presumably form a similar homodimer. Indeed, recombinant WW2, but not WW1, of human SAV1 eluted as a dimer on the gel filtration column (*Figure 4—figure supplement 1D*). In addition, HA-SAV1 ΔSARAH interacted more strongly with FLAG-SAV1 WT than FLAG-SAV1 ΔWW1/2 and ΔWW2, from human cell lysates, confirming WW2-dependent self-association of SAV1 in vivo (*Figure 4—figure supplement 1E*). WW domains recognize proline-rich peptides through solvent-exposed hydrophobic residues (*Aragón et al., 2012*). SAV1 WW domains can bind to PPxY motifs of LATS1/2 (*Ni et al., 2015*; *Tapon et al., 2002*). However, the ligand-binding site of SAV1 WW2 is buried in the dimer, suggesting that LATS1/2 binding and dimer formation are competing events.

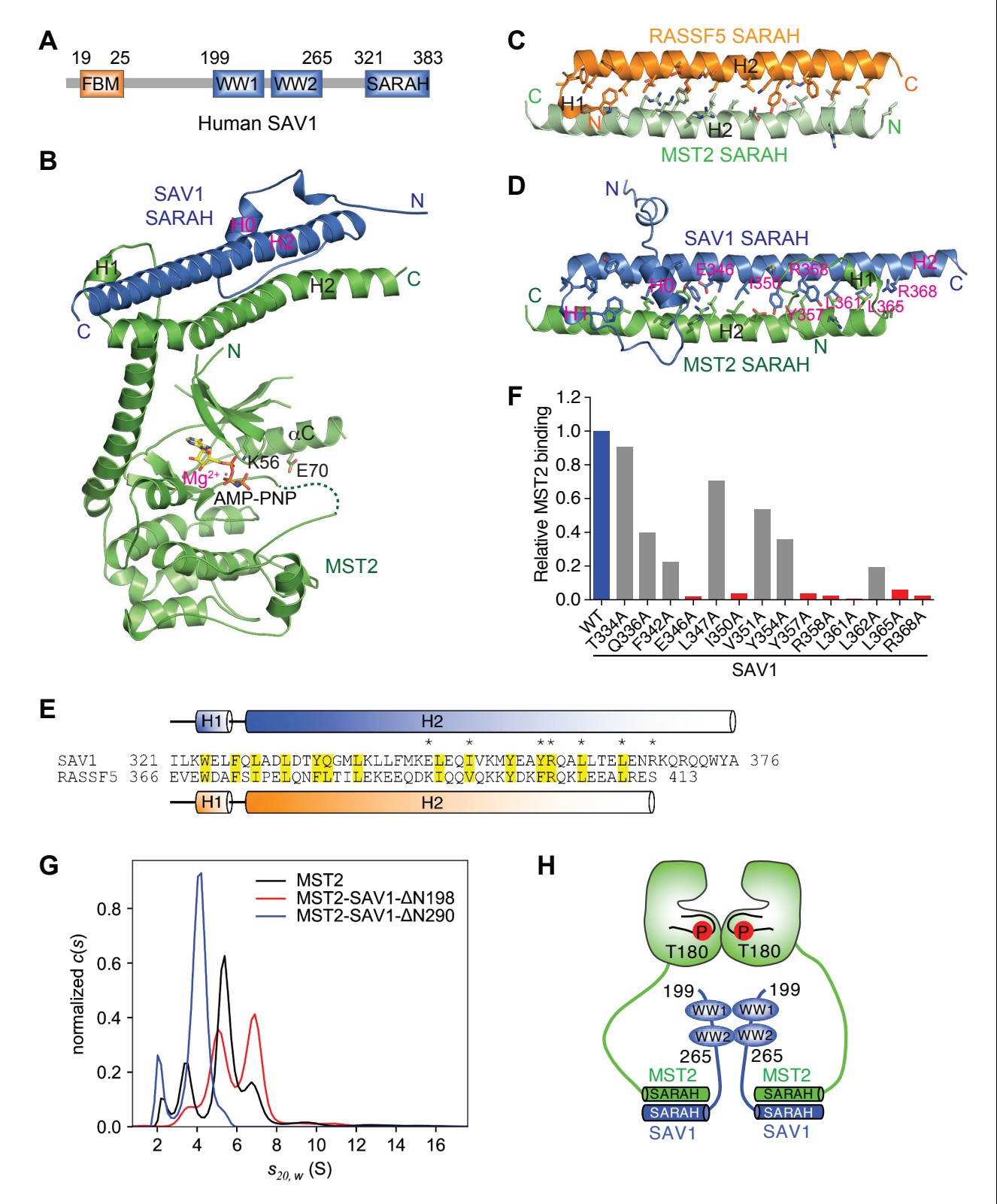

**Figure 4.** Crystal structure and binding interface of human MST2-SAV1. (**A**) Schematic drawing of domains and motifs of human SAV1. (**B**) Cartoon drawing of the crystal structure of the MST2-SAV1 complex. MST2 is colored green, and SAV1 is colored blue. Side chains of K56 and E70, and AMP-PNP are shown as sticks. Mg²⁺ is shown as a magenta sphere. The disordered T-loop is drawn as a green dashed line. (**C**) Cartoon drawing of the crystal structure of the MST2 SARAH-RASSF5 SARAH heterodimer (PDB ID: 4LGD). MST2 and RASSF5 are colored in light green and orange,
*Figure 4 continued on next page*

*Figure 4 continued*

respectively. The interface residues are shown as sticks. (**D**) Cartoon drawing of the crystal structure of the MST2 SARAH-SAV1 SARAH heterodimer (this study). MST2 and SAV1 are colored in green and blue, respectively. The interface residues are shown as sticks. SAV1 residues whose mutations cause defective MST2 binding are labeled in magenta. (**E**) Sequence alignment of SARAH domains of human SAV1 and RASSF5. The highly conserved interface residues are shaded in yellow. Seven interface residues whose mutations cause defective MST2 binding are marked by asterisks. The secondary structural elements of SAV1 SARAH are shown above the sequences and colored blue. The secondary structural elements of RASSF5 SARAH are shown below the sequences and colored orange. (**F**) Quantification of the relative binding intensity between MST2 SARAH domain (residue 431–491) and SAV1 SARAH domain (residue 321–383) and its mutants, derived from the pull-down experiments. Relative MST2 binding of the indicated SAV1 mutant is normalized against SAV1 wild type (WT; 100%). SAV1 mutants that lost or retained MST2 binding are colored red and gray, respectively. (**G**) Sedimentation velocity analytical ultracentrifugation analysis of MST2-FL (black), MST2-FL/SAV1-ΔN198 (red), and MST2-FL/SAV1-ΔN290 (blue). '$s_{20,w}$', sedimentation coefficient corrected to standard conditions; S, Svedberg units ($10^{-13}$ seconds). (**H**) Schematic model of the MST2-SAV1 heterotetramer.

DOI: https://doi.org/10.7554/eLife.30278.010

The following figure supplement is available for figure 4:

**Figure supplement 1.** SARAH and WW domains of SAV1 mediate the formation of the MST2-SAV1 heterotetramer.
DOI: https://doi.org/10.7554/eLife.30278.011

## All discernable domains of SAV1 are required for MST2 activation

Next, we further explored the mechanism by which SAV1 promoted MST2 activation in human cells. Overexpression of SAV1, but not RASSF1A (a RASSF1 isoform), enhanced MST2 T180 phosphorylation in un-stimulated cells (*Figure 5A*), indicating that SARAH domain-mediated MST2 binding alone is not sufficient to activate MST2. The MST2 3TA mutant had much higher pT180 signals, because it could not be inhibited by STRIPAK^SLMAP (*Figure 5B*). SAV1 strongly elevated the pT180 signal of MST2 WT, but did not further enhance T180 phosphorylation of MST2 3TA, suggesting that SAV1 stimulated MST2 activation through antagonizing STRIPAK^SLMAP in human cells.

We next examined which region of SAV1 contributed to MST2 activation. SAV1 ΔSARAH lost its ability to activate MST2 (*Figure 5C*). Mutations in SAV1 SARAH that disrupted its binding to MST2 (E346A/R358A and E346A/R368A) also diminished SAV1 stimulation of MST2 activation. Thus, SARAH-mediated binding of SAV1 to MST2 is required for SAV1-induced MST2 activation. Furthermore, deletion of the WW domains also decreased MST2 activation, with the deletion of WW2 having a larger effect, suggesting that WW2-mediated dimerization of SAV1 might be important for MST activation (*Figure 5D*). We have previously shown that SARAH-mediated MST2 homodimerization is critical for its activation (*Ni et al., 2013*). SAV1 SARAH disrupts this homodimerization of MST2 SARAH, and is expected to block MST2 activation. Indeed, SAV1 SARAH alone (SAV1 ΔN268) blocked MST2 activation in vitro (*Figure 5E*). Interestingly, the SAV1 ΔN198 mutant, which contains WW1/2 and SARAH domains, could support MST2 auto-activation in the same assay. These results suggest that SAV1 WW2 promotes the self-association of MST2–SAV1 heterodimers and enables trans-autophosphorylation and activation of the MST2 kinase domain in the context of the MST2–SAV1 heterotetramer (*Figure 4H*).

The SAV1 ΔN198 mutant did not enhance T180 phosphorylation of MST2 in human cells (*Figure 5F*), indicating that the N-terminal region of SAV1 is also important for MST2 activation. We made additional N-terminal truncation mutants of SAV1 to further map the region involved in MST2 activation. Deletion of the N-terminal 90 residues completely abolished MST2 pT180 signals, whereas deletion of the N-terminal 30 residues greatly reduced MST2 pT180. Because SAV1 1–30 contains the FBM motif that binds to the FERM domain of NF2 (*Yu et al., 2010*), we tested if SAV1 FBM was required for MST2 activation. The FBM deletion mutant of SAV1 was indeed deficient in MST2 activation (*Figure 5—figure supplement 1*), raising the intriguing possibility that NF2 or other FERM-containing proteins might regulate the ability of SAV1 to stimulate MST2.

Overexpression of SAV1 not only promoted MST2 T180 phosphorylation, but also increased phosphorylation at T336 and T378, which was required for SLMAP binding (*Figure 5G*). Consistent with this finding, SAV1 overexpression did not reduce the binding of SLMAP and STRIP1 to MST2 (*Figure 5H*). Collectively, these results suggest that SAV1 prevents pT180 dephosphorylation without disrupting the interaction between MST2 and STRIPAK^SLMAP. All discernable domains of SAV1 are required for MST2 activation in vivo.

**Table 3.** Data collection and refinement statistics for the MST2–SAV1 complex.

**Data collection**

| | |
|---|---|
| Crystal | Native |
| Space group | R32 |
| Wavelength (Å) | 0.97918 |
| Cell dimensions | |
| $a$, $b$, $c$ (Å) | 223.68, 223.68, 79.65 |
| $\alpha$, $\beta$, $\gamma$ (°) | 90.00, 90.00, 120.00 |
| Resolution range (Å) | 42.27–2.95 (3.00–2.95)[*] |
| Unique reflections | 15,976 (763) |
| Multiplicity | 19.0 (12.0) |
| Data completeness (%) | 99.9 (98.5) |
| $R_{merge}$ (%)[†] | 12.4 (171.5) |
| $R_{pim}$ (%)[‡] | 2.9 (48.3) |
| I/$\sigma$(I) | 28.3 (1.0) |
| $CC_{1/2}$[§] | 0.841 |
| Wilson $B$-value (Å$^2$) | 48.5 |
| Refinement Statistics | |
| Resolution range (Å) | 42.27–2.95 (3.18–2.95) |
| No. of reflections $R_{work}$/$R_{free}$ | 13,199/650 (1,027/52) |
| Data completeness (%) | 82.3 (34.0) |
| Atoms (non-H protein/solvent/metal) | 3,340/31/1 |
| $R_{work}$ (%) | 23.5 (32.2) |
| $R_{free}$ (%) | 25.2 (27.9) |
| R.m.s.d. bond length (Å) | 0.007 |
| R.m.s.d. bond angle (°) | 0.600 |
| Mean $B$-value (Å$^2$) (protein/solvent/ions) | 55.4/43.4/33.7 |
| Ramachandran plot (%) (favored/additional/disallowed)[#] | 94.2/4.5/1.3 |

[*]Data for the highest-resolution shell are shown in parentheses.

[†]$R_{merge}$ = 100 $\Sigma_h \Sigma_i |I_{h,i} - \langle I_h \rangle|/\Sigma_h \Sigma_i \langle I_{h,i} \rangle$, where the outer sum (h) is over the unique reflections and the inner sum (i) is over the set of independent observations of each unique reflection.

[‡]$R_{pim}$ = 100 $\Sigma_h \Sigma_i [1/(n_h - 1)]^{1/2} |I_{h,i} - \langle I_h \rangle|/\Sigma_h \Sigma_i \langle I_{h,i} \rangle$, where $n_h$ is the number of observations of reflections h.

[§]$CC_{1/2}$ values shown are for the highest resolution shell.

[#]As defined by the validation suite MolProbity.

DOI: https://doi.org/10.7554/eLife.30278.012

One possibility is that SAV1 may target MST2 to the plasma membrane, thus spatially separating MST2 from STRIPAK[SLMAP], whose SLMAP subunit has been reported to localize to the endoplasmic reticulum (*Guzzo et al., 2004*; *Hwang and Pallas, 2014*; *Nader et al., 2012*; *Wigle et al., 1997*). To test this possibility, we examined the localization of MST2 and SAV1 in 293FT cells by subcellular fractionation (*Figure 5—figure supplement 2*). Tubulin, NF2 and Pan Cadherin were used as markers for the cytosolic, peripheral and integral membrane fractions, respectively. The membrane fraction contained both peripheral and integral membrane proteins. Although a minor fraction of SAV1 and MST2 resided in the membrane fraction, major pools of SAV1 and MST2 were present in the cytosol. More importantly, both cytosolic and membrane fractions of MST2 were proportionally activated, as indicated by phosphorylation at T180. Thus, membrane targeting cannot fully explain SAV1-dependent activation of MST2. SAV1 may directly antagonize STRIPAK-mediated inhibition of MST2. Because Hippo/MST binding can induce SAV phosphorylation and prevent its degradation, leading to increased levels of SAV in cells (*Aerne et al., 2015*; *Callus et al., 2006*; *Pantalacci et al.,*

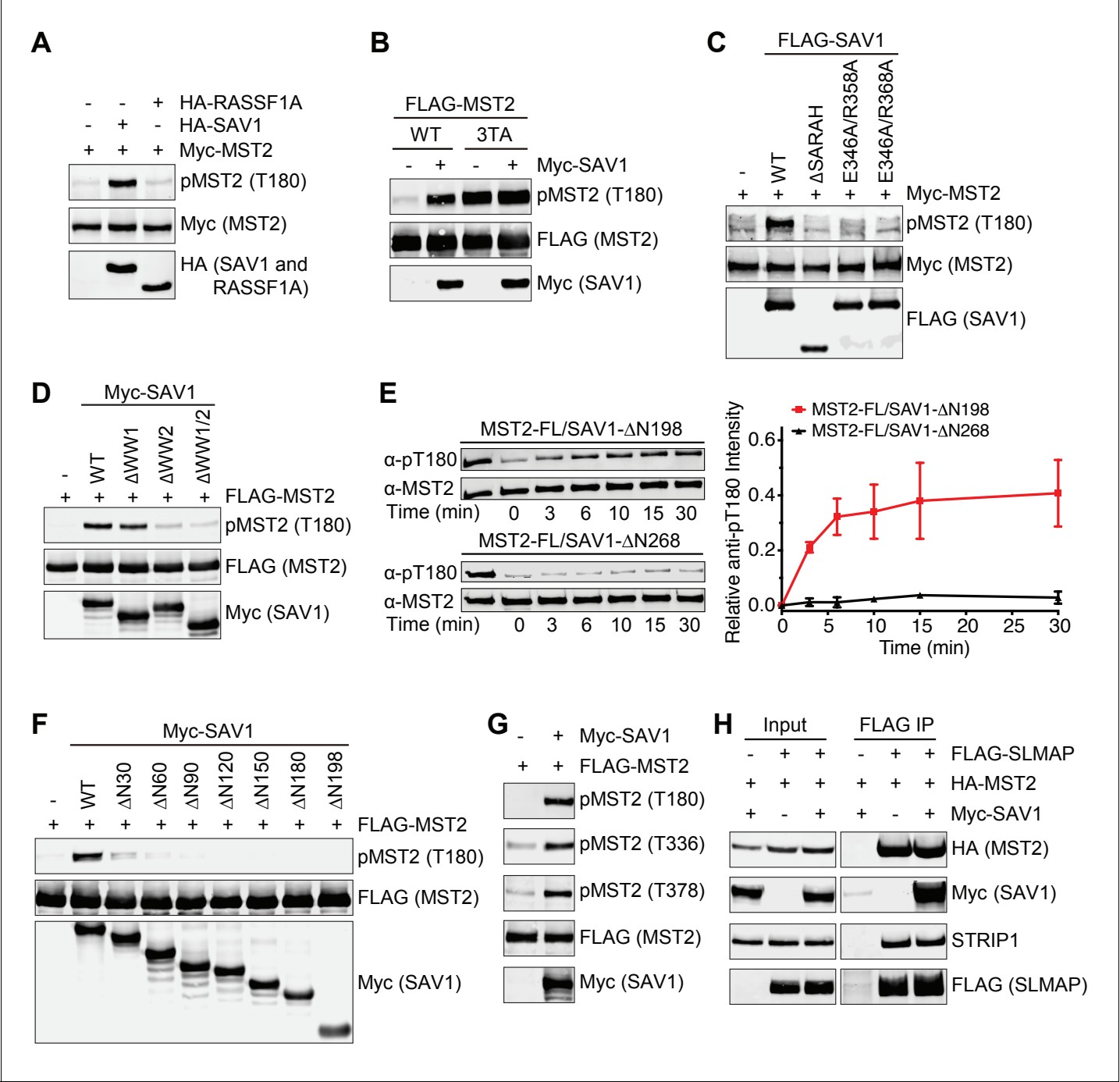

**Figure 5.** All discernable domains of SAV1 are required for MST2 activation. (**A**) Immunoblots of lysates of 293FT cells co-transfected with the indicated Myc-MST2 and HA-SAV1 or HA-RASSF1A plasmids. (**B**) Immunoblots of lysates of 293FT cells co-transfected with Myc-SAV1 and the indicated FLAG-MST2 plasmids. (**C**) Immunoblots of lysates of 293FT cells co-transfected with Myc-MST2 and the indicated FLAG-SAV1 plasmids. (**D**) Immunoblots of cell lysates of 293FT cells co-transfected with FLAG-MST2 and the indicated Myc-SAV1 plasmids. (**E**) Total MST2 and pT180 blots of the auto-kinase reactions by the indicated MST2-SAV1 complexes at the indicated time points (left panel). The kinetic profiles of T180 autophosphorylation of MST2-FL/SAV1-ΔN198 (red line) and MST2-FL/SAV1-ΔN268 (black line) are shown on the right. The relative anti-pT180 intensities were normalized against MST2-FL without PP2A phosphatase treatment (100%). Data are representative of at least two independent experiments. (**F**) Immunoblots of lysates of 293FT cells co-transfected with FLAG-MST2 and the indicated N-terminal truncation of Myc-SAV1 constructs. (**G**) 293FT cells were co-transfected with FLAG-MST2 and Myc-SAV1 plasmids. The total cell lysates were blotted with the indicated antibodies. (**H**) 293FT cells were co-transfected with the indicated plasmids encoding FLAG-SLMAP, HA-MST2 and Myc-SAV1. Anti-FLAG IP and the total cell lysates (input) were blotted with the indicated antibodies.
DOI: https://doi.org/10.7554/eLife.30278.013

*Figure 5 continued on next page*

*Figure 5 continued*

The following figure supplements are available for figure 5:

**Figure supplement 1.** The N-terminal region of SAV1 is required for MST2 activation.

DOI: https://doi.org/10.7554/eLife.30278.014

**Figure supplement 2.** SAV1 and MST2 are present in the cytosol.

DOI: https://doi.org/10.7554/eLife.30278.015

*2003*; *Wu and Wu, 2013*; *Wu et al., 2003*), SAV may also function in a positive feedback mechanism in Hippo/MST activation (*Aerne et al., 2015*).

## The N-terminal region of SAV1 directly inhibits PP2A

The N-terminal residues of SAV1 are required for MST2 activation in human cells, but are not expected to contact MST2. We hypothesized that this region of SAV1 might directly interact with PP2A A-C core and inhibit its phosphatase activity. The SAV1 1–198 fragment, which was incapable of binding to MST2, interacted with both PP2A A and C subunits in human cell lysates (*Figure 6A*). Consistent with the MST2 activation assay, deletion of the N-terminal 90 residues from this fragment abolished this interaction whereas deletion of the N-terminal 30 residues greatly weakened the interaction. Furthermore, the SAV1 1–90 fragment, which did not associate with MST1/2 or SLMAP (*Figure 6B* and *Figure 6—figure supplement 1*), also bound to PP2A A subunit in human cell lysates. Expectedly, SAV1 ΔN90 interacted with MST2 through the SARAH domain. Its interaction with PP2A A subunit was likely bridged through MST2, SLMAP, and other components of STRIPAK. These results suggest that the N-terminal 90 residues of SAV1 might possess certain affinity for the catalytic core of PP2A, independent of MST2 and SLMAP.

Recombinant SAV1 1–90 co-fractionated with recombinant PP2A A subunit on a gel filtration column (*Figure 6C*), consistent with the formation of a SAV1–PP2A complex. More importantly, SAV1 1–90 blocked the dephosphorylation of MST2 pT180 by PP2A A-C in vitro (*Figure 6D*). Therefore, the N-terminal region of SAV1 can directly inhibit the phosphatase activity of PP2A.

## SAV1 stimulates MST1/2 activation through antagonizing STRIPAK

Our results so far strongly suggest that SAV1 activates MST1/2 by directly counteracting STRIPAK[SLMAP]-mediated suppression of MST1/2 activation. Very high and non-physiological concentrations of SAV1 1–90 (100 μM) are needed for PP2A A-C inhibition, indicating that SAV1 cannot inhibit all forms of PP2A. Instead, it likely inhibits the PP2A A-C core in the context of the SAV1–MST2–STRIPAK[SLMAP] complex. Other SAV1 elements, such as the SARAH and WW domains, may position the N-terminal region of SAV1 for optimal inhibition of PP2A.

To test this possibility, we examined whether SAV1, when bound to MST2, could interact with STRIPAK. When co-expressed with MST2 WT in 293FT cells, SAV1 bound to SLMAP and several core components of STRIPAK, including PP2A A-C, SIKE1, and STRIP1 (*Figure 7A*), suggesting that MST2–SAV1 could form a complex with STRIPAK[SLMAP]. Consistent with a requirement for the phospho-MST2 linker in the assembly of the specific SAV1–MST2–STRIPAK[SLMAP] complex, MST2 7TA, which was deficient in SLMAP binding, did not support the binding of SAV1 to SLMAP, SIKE, or STRIP1. When co-expressed with MST2 7TA, SAV1 still interacted with PP2A A-C, albeit more weakly, consistent with a direct, transient interaction between the N-terminal region of SAV1 and the PP2A A-C catalytic core.

If a major mechanism by which SAV1 promotes MST2 activation is through inhibiting STRIPAK[SLMAP]-mediated dephosphorylation of MST2 pT180, then deletion of SLMAP is expected to bypass the requirement for SAV1 in MST2 activation in vivo. We thus created 293FT cells with both *SAV1* and *SLMAP* deleted (DKO), and examined the phosphorylation level of Hippo pathway components and the expression of Hippo target genes in un-stimulated cells (*Figure 7B and C*, and *Figure 7—figure supplement 1*). As shown in previous sections, SLMAP KO and SAV1 KO cells had opposite effects on the phosphorylation level of Hippo pathway components and the expression of Hippo target genes, *CTGF* and *CYR61*, respectively, consistent with their opposing roles in the Hippo pathway. Strikingly, SLMAP/SAV1 DKO cells exhibited a very similar phenotype as that of SLMAP KO cells. Thus, deletion of SLMAP bypasses the function of SAV1 in Hippo signaling at least in un-

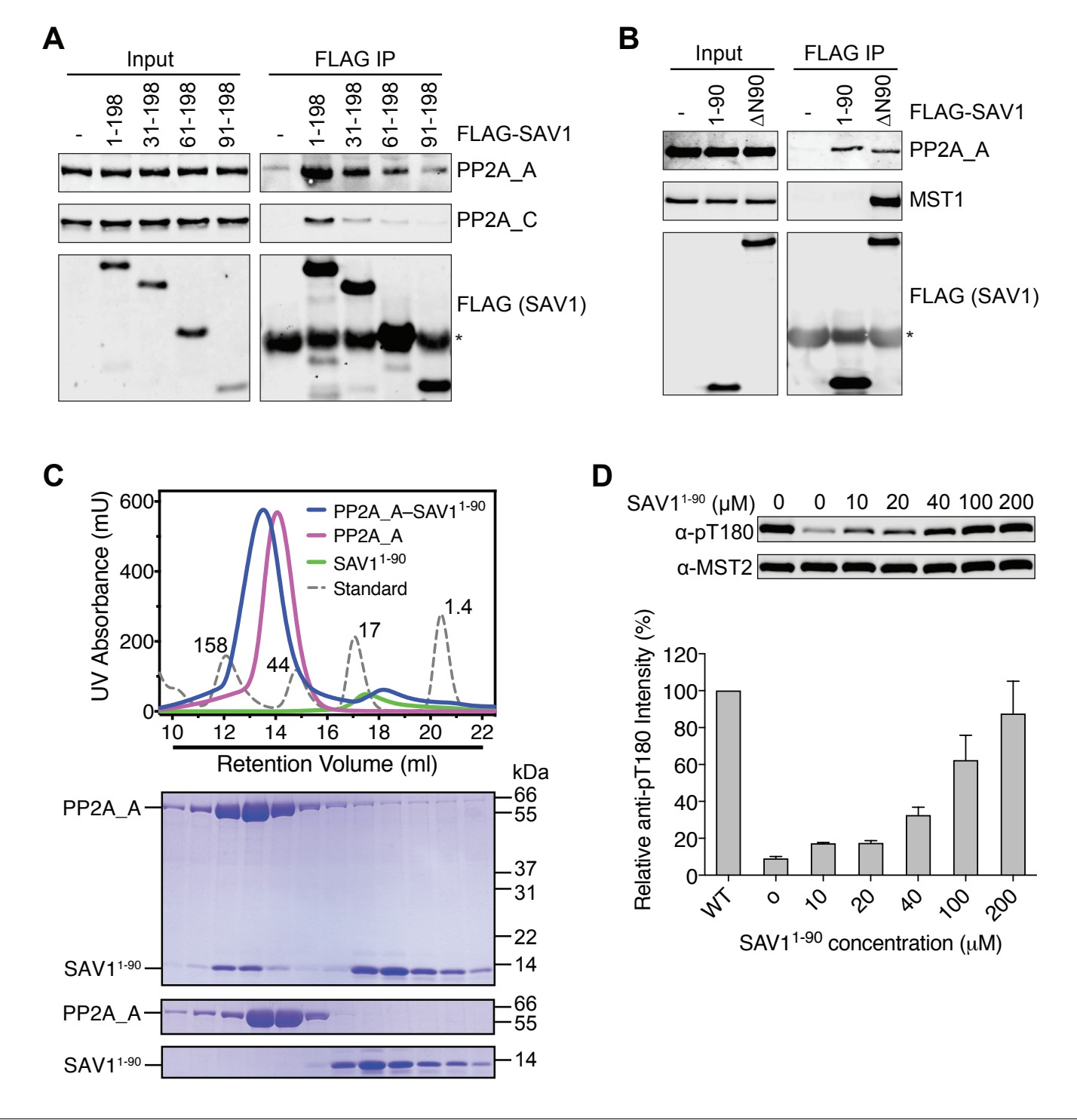

**Figure 6.** The N-terminal region of SAV1 directly inhibits PP2A. (**A and B**) Association of PP2A A-C with the N terminal region of SAV1. 293FT cells were mock transfected or transfected with the indicated FLAG-SAV1 plasmids. The total cell lysates (input) and anti-FLAG IP were blotted with the indicated antibodies. Asterisk designates IgG. (**C**) UV traces of the PP2A A and SAV1$^{1-90}$ complex (blue line), PP2A A alone (magenta line), SAV1$^{1-90}$ alone (green line), and molecular weight standard (dashed line, corresponding molecular weight indicated in kDa) fractionated on a Superdex 200 gel filtration column, respectively. The underlined fractions were separated on SDS-PAGE and stained with Coomassie. (**D**) The N-terminal region of SAV1 directly inhibits dephosphorylation of MST2 pT180 by PP2A. Quantitative MST2 pT180 and total MST2 immunoblotting of phosphatase reactions containing human MST2 and SAV1$^{1-90}$ at the indicated concentrations. The relative anti-pT180 intensities were normalized against MST2 without PP2A phosphatase treatment (100%). Data are representative of at least two independent experiments.

*Figure 6 continued on next page*

*Figure 6 continued*

DOI: https://doi.org/10.7554/eLife.30278.016

The following figure supplement is available for figure 6:

**Figure supplement 1.** The N-terminal 90 residues of SAV1 associates with PP2A_A subunit in human cells.
DOI: https://doi.org/10.7554/eLife.30278.017

stimulated cells. These results strongly suggest that a major function of SAV1 in the Hippo pathway is to counteract the suppression of MST activation by STRIPAK[SLMAP].

## Discussion

The Hippo pathway is a critical signaling pathway that controls tissue growth and homeostasis. Diverse upstream signals activate the central MST-LATS kinase cascade in this pathway, which regulates the YAP/TAZ-TEAD transcription module to alter transcription. Among the four core components of the MST-LATS kinase cascade (MST, SAV1, LATS, and MOB1), the biochemical function of SAV1 is the least understood. Our present study has filled this important gap, and establishes a molecular framework for SAV1-dependent MST activation (*Figure 7D*).

MST2 undergoes constitutive trans-autophosphorylation at T180 and several residues in its linker. Multiple phospho-residues in the MST2 linker redundantly bind to the FHA domain of SLMAP, an adaptor of STRIPAK, and promote PP2A-mediated dephosphorylation of pT180, resulting in feedback inactivation of MST2 under normal growth conditions. Upon activation of the Hippo pathway by contact inhibition or cytoskeletal perturbation, SAV1 binds to MST2 and forms a SAV1–MST2 heterodimer through their SARAH domains. SAV1 WW domains further dimerize to promote the formation of a SAV1-MST heterotetramer. In the context of the MST2-SAV1 heterotetramer, the two kinase domains of MST2 can still undergo trans-autophosphorylation at T180 and activation. The molecular interactions among MST2, SAV1, and STRIPAK[SLMAP] position the N-terminal phosphatase-inhibitory domain (PID) of SAV1 for direct binding to and inhibition of the PP2A catalytic core, thus protecting pT180 from dephosphorylation. Therefore, SAV1 promotes MST1/2 activation through forming an MST1/2-SAV1 heterotetramer that permits MST1/2 autoactivation, and through antagonizing PP2A-mediated dephosphorylation of the activation loop of MST1/2. Although the N-terminal PID of SAV1 in isolation interacts with PP2A weakly and only inhibits PP2A A/C at very high concentrations, this domain may be more effective in PP2A inhibition in the context of the intact SAV1–MST2–STRIPAK[SLMAP] complex. Scaffolding by striatin and other adaptor proteins in this complex may buttress the interaction of SAV1 PID and PP2A and position this domain for optimal PP2A inhibition. Future structural studies on the SAV1–MST2–STRIPAK[SLMAP] super-complex will definitively test the key tenets of this framework.

In *Drosophila*, dRASSF has been reported to promote the association between dSTRIPAK and Hippo, suggesting that dRASSF may also help to recruit dSTRIPAK to inactivate Hippo by dephosphorylation of its activation loop (*Ribeiro et al., 2010*). Therefore, binding to RASSF could provide an alternative mechanism for STRIPAK recruitment to inactivate MST1/2 and block Hippo signaling.

We have previously shown that MOB1 is an important downstream substrate of MST1/2. Phosphorylation of MOB1 by MST1/2 requires the binding of MOB1 to phospho-residues in the MST1/2 linker. MOB1 can thus compete with SLMAP for binding to overlapping sets of phospho-residues and further enhance MST1/2 activation in a feed-forward mechanism.

SAV1 has been proposed to perform additional functions in the Hippo pathway. For example, MST1/2 can be targeted to the plasma membrane by SAV1, whereas NF2 promotes the plasma membrane association of LATS1 (*Yin et al., 2013*). The co-localization of MST1/2 and LATS1/2 at the membrane facilitates MST1/2-dependent phosphorylation and activation of LATS1/2. In addition, SAV1 has been proposed to recruit substrates to MST1/2 for efficient phosphorylation (*Pantalacci et al., 2003*; *Tapon et al., 2002*; *Udan et al., 2003*; *Wu et al., 2003*). Our discovery that SAV1 antagonizes STRIPAK[SLMAP] to promote MST1/2 activation is entirely consistent with these observations, and further implies that SAV1 targets the active MST1/2 to appropriate substrates. Deletion of SLMAP bypasses the requirement of SAV1 in Hippo pathway activation in normal growth conditions, providing conclusive evidence for the antagonism between SAV1 and STRIPAK[SLMAP]in

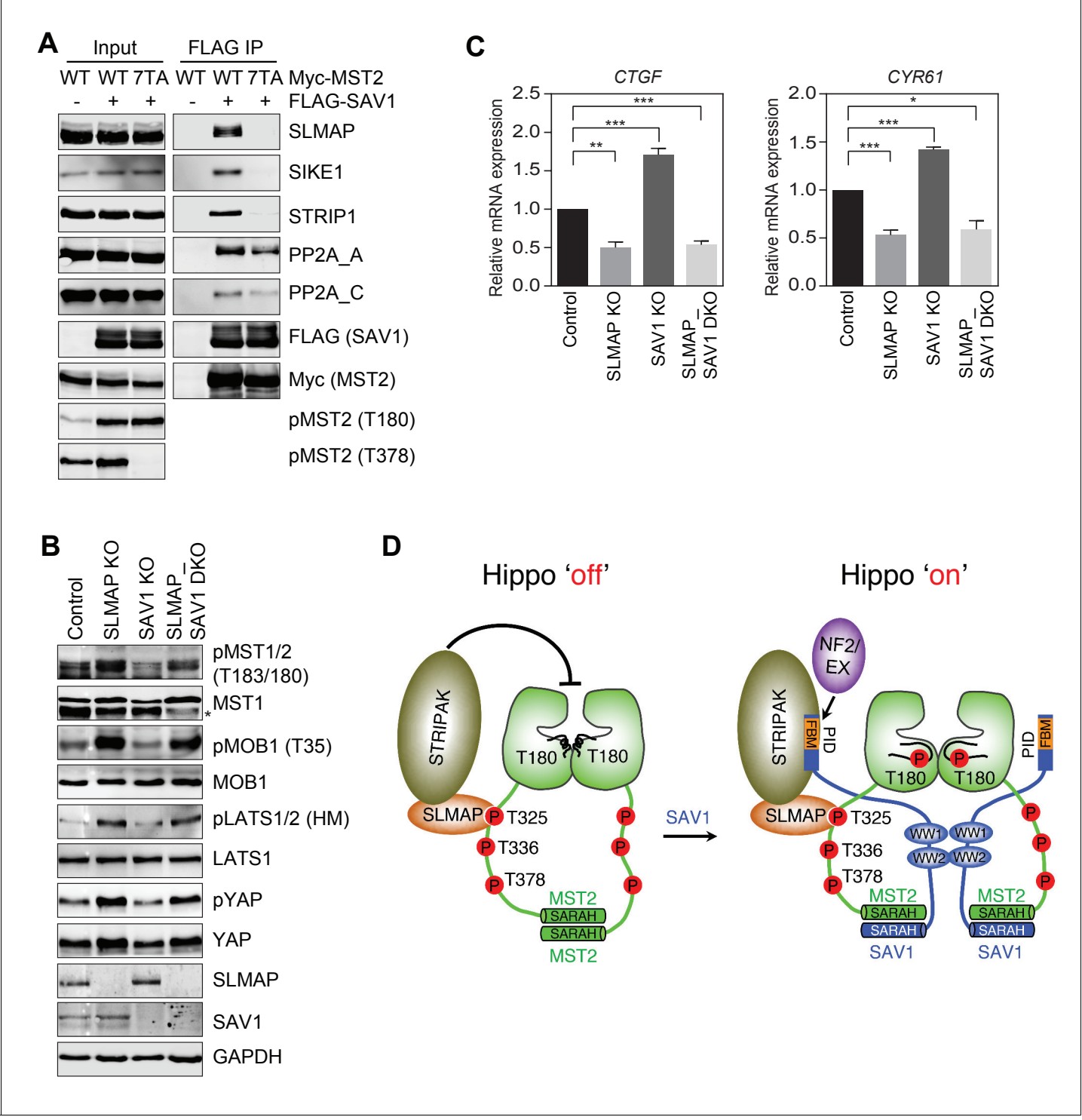

**Figure 7.** SAV1 stimulates MST1/2 activation through antagonizing STRIPAK. (A) SAV1-MST2 forms a complex with STRIPAK^SLMAP. 293FT cells were co-transfected with FLAG-SAV1 and the indicated Myc-MST2 plasmids. The total cell lysates (input) and anti-FLAG IP were blotted with the indicated antibodies. (B) Immunoblots with the indicated antibodies of lysates of control MCF10A cells and MCF10A cells with *SLMAP*, *SAV1*, or both deleted. Asterisk denotes a non-specific band. (C) Relative expression of YAP target genes *CTGF* and *CYR61* in control, SLMAP KO, SAV1 KO, and SLMAP/SAV1 DKO MCF10A cells. The gene expression was measured by quantitative real-time RT-PCR and normalized to GAPDH. Data are plotted as mean ± SEM of three biological replicates (*p<0.05; **p<0.01; and ***p<0.001). (D) Model for SAV1-dependent MST2 activation during Hippo signaling.
DOI: https://doi.org/10.7554/eLife.30278.018

The following figure supplement is available for figure 7:

*Figure 7 continued on next page*

*Figure 7 continued*

**Figure supplement 1.** CRISPR/Cas9-induced indel mutations in MCF10A and 293FT cells.

DOI: https://doi.org/10.7554/eLife.30278.019

vivo. On the other hand, it is unlikely that all SAV1 functions can be bypassed by SLMAP deletion, given the wide array of upstream stimuli that can activate the Hippo pathway.

The Hippo pathway can be activated by a multitude of signals and stimuli. The SAV1–STRIPAK$^{SLMAP}$ antagonism is likely regulated by upstream signals. STRIPAK$^{SLMAP}$ is expected to outcompete the inhibitory function of SAV1 when the Hippo pathway is off, and the opposite might be true when the Hippo pathway is on. Currently, we do not know how the SAV1– STRIPAK$^{SLMAP}$ antagonism is regulated by upstream signals to affect Hippo kinase activation. Along this vein, the N-terminal PID of SAV1 encompasses the FBM that binds to FERM domains. Deletion of SAV1 FBM reduces its ability to activate MST2 in human cells. SAV1 PID directly binds to and inhibits the PP2A catalytic core, thus protecting MST2 from dephosphorylation. Because the interaction between SAV1 PID and PP2A A/C in isolation is weak, it may need to be buttressed by scaffolding functions of other components in the STRIPAK$^{SLMAP}$–MST2–SAV1 complex. In another non-exclusive possibility, through the embedded FBM, SAV1 PID may further recruit certain FERM-containing proteins, such as NF2 and Expanded (EX), to promote STRIPAK inhibition in vivo. In this case, NF2 or EX might directly bind to the core components of STRIPAK$^{SLMAP}$. Through engaging both SAV1 PID and STRIPAK, NF2 or EX might help to position SAV1 PID in the optimal orientation to inhibit STRIPAK during Hippo signaling activation. Intriguingly, when the Hippo pathway is off, NF2 adopts a closed conformation, in which its FERM domain is autoinhibited and cannot bind to FBMs (*Li et al., 2015*). When the Hippo pathway is on, NF2 adopts an open conformation, in which its FERM domain binds to the FBM of LATS1/2 and recruits LATS1/2 to membrane for phosphorylation by MST1/2. Oncogenic NF2 mutations are believed to alter the open-closed conformational change of NF2, thus inhibiting Hippo signaling. It will be interesting to test whether NF2 or other FERM-containing proteins, such as EX, regulate the PP2A-blocking function of SAV1 in similar ways.

A major unresolved question is how the Hippo kinases are activated by upstream signals. In this study, we have shown that the dynamic antagonism between SAV1 and STRIPAK determines the activation status of MST1/2. By virtue of its ability to interact with NF2/EX, STRIPAK, and MST1/2, SAV1 links upstream signals to MST1/2 activation. Because STRIPAK contains both a kinase and a phosphatase, the delicate balance between the opposite catalytic activities of the two enzymes in the same complex can play a critical role in toggling the activation status of Hippo signaling. Although the Hippo pathway is often dysfunctional in human cancer, the core components of the MST-LATS kinase cascade are rarely mutated (*Harvey et al., 2013*). Chemical inhibitors of the STRIPAK complex are expected to activate MST1/2 and downstream signaling, and may have therapeutic potential in treating human cancers driven by NF2 mutations or elevated expression of YAP/TAZ.

## Materials and methods

### Protein expression and purification

The coding regions of human SLMAP FHA (residues 1–140), the C-terminal fragments of SAV1, and PP2A A subunit were cloned into a modified pGEX-6P vector (GE Healthcare) that included a tobacco etch virus (TEV) protease cleavage site at the N terminus, respectively. The coding regions of human MST2 and the N-terminal fragment of SAV1 (residues 1–90) were cloned into a modified pET29 vector (EMD Millipore) that included an N-terminal His$_6$ tag, respectively. Mutations were generated using QuickChange II XL site-directed mutagenesis kit (Agilent Technologies), and constructs were verified by DNA sequencing. All plasmids were transformed into the bacterial strain BL21 (DE3)-T1$^R$ cells (Sigma) for protein expression. SLMAP FHA wild-type (WT) and corresponding mutants were purified with glutathione-Sepharose beads (GE Healthcare) and cleaved with TEV overnight at 4°C to remove the GST moiety. The cleaved SLMAP FHA proteins were further purified with a Superdex 75 size exclusion column (GE Healthcare) in a buffer containing 20 mM Tris (pH 8.0), 100 mM NaCl and 1 mM TCEP. Co-expressed MST2 and the C-terminal fragments of SAV1 were purified with glutathione-Sepharose beads and cleaved with TEV overnight at 4°C to remove the GST moiety.

The MST2-SAV1 complex was further purified by a Superdex 200 size-exclusion column (GE Health-care) equilibrated with a buffer containing 20 mM Tris (pH 8.0), 200 mM NaCl, 5 mM $MgCl_2$, and 1 mM TCEP. PP2A A subunit was purified with glutathione-Sepharose beads, cleaved with TEV over-night at 4°C to remove the GST moiety, and further purified by a Superdex 200 column with a buffer containing 20 mM Tris (pH 8.0), 100 mM NaCl, and 1 mM TCEP.

The N-terminal fragment of SAV1$^{1-90}$ was purified by denaturation method using 8 M urea according to the manufacturer's instructions (Qiagen). SAV1$^{1-90}$ was refolded overnight at 4°C in a buffer containing 50 mM Tris (pH 7.5), 150 mM NaCl, 5 mM $MgCl_2$, and 1 mM DTT. To obtain the PP2A A and SAV1$^{1-90}$ complex, PP2A A was mixed with refolded SAV1$^{1-90}$ at a 1:6 molar ratio and further purified by a Superdex 200 column equilibrated with a buffer containing 20 mM Tris (pH 8.0), 50 mM NaCl, and 1 mM TCEP. For comparison, PP2A A and SAV1$^{1-90}$ alone was run under the same condition, respectively.

## Crystallization, data collection, and structure determination

SLMAP FHA (residue 1–140) was concentrated to 12 mg/ml and crystallized at 20°C using the sit-ting-drop vapor diffusion method with a reservoir solution containing 100 mM Bis-Tris (pH 6.5), 200 mM $Li_2SO_4$, and 25% (w/v) PEG 3350. The crystals were cryoprotected with the reservoir solution supplemented with 25% (v/v) glycerol and then flash-cooled in liquid nitrogen. Crystals of apo SLMAP FHA diffracted to a resolution of 1.08 Å and contained one molecule per asymmetric unit. SLMAP FHA was concentrated to 23.7 mg/ml and mixed with the pMST2 peptide (EEEDGpTMKRN) to a 1:2 molar ratio. SLMAP FHA-pMST2 was crystallized at 20°C using the hanging drop vapor diffu-sion method with a reservoir solution containing 200 mM KSCN and 20% (w/v) PEG3350. The crys-tals were cryoprotected with the reservoir solution supplemented with 30% (v/v) glycerol. Crystals of SLMAP FHA–pMST2 diffracted to a resolution of 1.55 Å and contained two SLMAP FHA–pMST2 molecules per asymmetric unit. The MST2 ΔL/D146N–SAV1 SARAH complex was concentrated to 8 mg/ml and crystallized at 20 °C using the hanging-drop vapor-diffusion method with a reservoir solu-tion containing 0.1 M HEPES (pH 7.5), 0.19 mM CYMAL-7, and 40% (v/v) PEG 400. The crystals were then flash-cooled in liquid nitrogen. Native crystals diffracted to a minimum Bragg spacing ($d_{min}$) of 2.95 Å and exhibited the symmetry of space group R32 with cell dimensions of a = 223.7 Å, c = 79.7 Å and contained one MST2-SAV1 complex per asymmetric unit.

All diffraction data were collected at beamline 19-ID (SBC-CAT) at the Advanced Photon Source (Argonne National Laboratory, Argonne, Illinois, USA) and processed in the program HKL-3000 (*Minor et al., 2006*). Initial phases for the apo SLMAP FHA structure were obtained by molecular replacement in the program Phaser (*McCoy et al., 2007*) using the structure of the FHA domain of human centrosomal protein CEP170 as a search model (PDB ID: 4JON). Initial phases for SLMAP FHA with pMST2 were obtained by molecular replacement in the program Phaser using the structure of apo SLMAP FHA as a search model. Iterative model building and refinement were performed in the programs Coot (*Emsley et al., 2010*) and Phenix (*Adams et al., 2010*), respectively. Data collec-tion and refinement statistics are summarized in *Table 2*.

MST2 ΔL/D146N–SAV1 SARAH data were processed in the program HKL-3000 with applied cor-rections for effects resulting from absorption in a crystal and for radiation damage (*Borek et al., 2003*; *Otwinowski et al., 2003*), the calculation of an optimal error model, and corrections to com-pensate the phasing signal for a radiation-induced increase of non-isomorphism within the crystal (*Borek et al., 2010*; *Borek et al., 2013*). These corrections were crucial for successful phasing. Crys-tals of MST2 ΔL/D146N–SAV1 SARAH displayed anisotropic diffraction and while 100% complete to 3.2 Å resolution, rapidly fell off in intensity to the high-resolution limit of 2.95 Å resolution. To maxi-mize completeness for the final data set, data from three isomorphous crystals were merged for the equivalence of a sweep of 325°. Phases for MST2 ΔL/D146N–SAV1 SARAH were obtained by molec-ular replacement in the program Phaser. The crystal structure of human MST2 ΔL/D146N–RASSF5 SARAH (PDB ID: 4LGD) (*Ni et al., 2013*) was modified for use as a search model by converting the RASSF5 SARAH polypeptide to a poly-alanine chain that was truncated to the minimal 37 residues of the SARAH helix (residues 376–413). Density modification in the program Parrot (*Cowtan, 2010*) and model building in the program Buccaneer (*Cowtan, 2006*) yielded a model for the SAV1 SARAH with 73 residues (78%) built and primary sequence assigned; confirmation of the sequence of SAV1 SARAH was determined by packing interactions with the MST2 ΔL/D146N SARAH domain and omit maps calculated after rounds of model refinement in the program Phenix. Completion of this model

was performed by manual rebuilding in the program Coot. Refinement was performed in the program Phenix with reference model restraints for the MST2 kinase domain supplied from monomer A of the high resolution MST2 KD/D146N domain, truncated at the C-terminus to residue 299 (PDB ID: 4LG4) (*Ni et al., 2013*). The final model for MST2 ΔL/D146N–SAV1 SARAH ($R_{work}$ = 22.9%, $R_{free}$ = 25.4%) contained 424 residues and one molecule of $Mg^{2+}$-AMP-PNP. Five residues are outliers in a Ramachandran plot as defined in the program MolProbity (*Chen et al., 2010*); all are located in MST2 ΔL/D146N, in surface loops with poor electron density. Data collection and structure refinement statistics are summarized in *Table 3*.

## In vitro binding assays

To assay the interactions between SLMAP FHA and MST2, GST-SLMAP FHA fusion protein was bound to glutathione-Sepharose beads as bait and incubated with purified MST2-FL, MST2-KD, and MST2-ΔL in TBS (50 mM Tris, pH 7.5, 150 mM NaCl) containing 0.05% (v/v) Tween for two hours. The beads were washed three times with TBS. The proteins retained on the beads were analyzed by SDS-PAGE.

To assay the heterodimerization of SARAH domains between MST2 and SAV1, Myc-tagged SAV1 SARAH (residues 321–383) and the corresponding mutants were in vitro translated in reticulocyte lysate in the presence of $^{35}$S-methionine according to the manufacturer's protocol (Pierce). Purified GST-MST2 SARAH (residues 430–491) fusion protein was bound to glutathione-Sepharose beads and incubated with $^{35}$S-labeled SAV1 SARAH WT and mutants. The beads were washed three times with TBS containing 0.05% (v/v) Tween. The proteins retained on the beads were analyzed by SDS-PAGE followed by autoradiography.

## Isothermal titration calorimetry (ITC)

The binding affinities between purified recombinant human SLMAP-FHA (residues 1–140) and the phospho-MST2 peptides were measured using a MicroCal iTC200 calorimeter (GE Healthcare) at 20 °C. The binding affinities between recombinant $MOB1^{33-216}$ and pT378-MST2 peptides were measured as controls. Ratio of protein to peptide concentration was 1:10 for each experiment. A 50 μM SLMAP-FHA or $MOB1^{33-216}$ sample in the buffer containing 20 mM Tris, pH 8.0, 50 mM NaCl was titrated with 500 μM of phospho-MST2 peptide in the same buffer. All phospho-MST2 peptides were synthesized with the following sequences: pT325-$MST2^{320-329}$, ELDSHpTMVKT; pT336-$MST2^{331-340}$, VESVGpTMRAT; pT378-$MST2^{373-382}$, EEEDGpTMKRN; and pT378-$MST2^{371-401}$, DEEEEDGpTMKRNATSPQVQRPSFMDYFDKQD. Due to low solubility of pT325-$MST2^{320-329}$, 5% (v/v) DMSO was added to both protein and peptide buffers. Peptide concentration was measured using the spectrophotometer at wavelength λ = 205 nm. In all cases, thermograms were integrated using NITPIC method (*Keller et al., 2012*), imported to the SEDPHAT software (*Zhao et al., 2015*), and binding-model fits (A + B < ->AB) were made by fixing the protein concentration adjustment coefficients and allowing the software to adjust peptide concentration by a correction coefficient, as it is not 100% accurate due to the nature of the concentration measurements. All ITC figures were rendered in GUSSI (*Brautigam, 2015*). Data are summarized in *Table 1*.

## Analytical ultracentrifugation (AUC)

All analytical ultracentrifugation was carried out at 4°C in an Optima XL-I centrifuge (Beckman-Coulter, Indianapolis, IN). The centrifugation cells comprised dual-sectored, charcoal-filled Epon centerpieces sandwiched between sapphire windows in standard aluminum housings. Approximately 390 μL of AUC Buffer (20 mM Tris pH 8.0, 200 mM NaCl, 5 mM $MgCl_2$) was placed into the reference sector, and the same volume of MST2 or MST2-SAV1 mixture was introduced into the sample sector. The cells were inserted into an An50Ti rotor and allowed to equilibrate in the centrifuge overnight under vacuum. Centrifugation was then initiated at 50,000 rpm. Concentration profiles were collected using the absorbance optics tuned to 280 nm. The data were analyzed using the $c(s)$ model in SEDFIT (*Schuck, 2000*) using a regularization level of 0.68 and an s resolution of 150; time-invariant noise was subtracted from the data (*Schuck and Demeler, 1999*). Values for partial-specific volume, buffer density, and buffer viscosity were calculated using SEDNTERP (*Laue et al., 1992*). AUC figures were rendered in GUSSI (*Brautigam, 2015*).

## In vitro kinase assays and autophosphorylation

For in vitro kinase assays, 0.4 µM MST2-FL or MST2-ΔL was incubated in the kinase reaction buffer containing 50 mM Tris (pH 7.5), 150 mM NaCl, 10 mM MgCl$_2$, 1 mM ATP, 1 mM DTT, and 0.1 µCi/µl γ-$^{32}$P-ATP with 40 µM myelin basic protein (MBP). At different time intervals, an aliquot of reaction mixture was added with equal volume of 2X SDS sample buffer and followed by boiling for 5 min in 100 °C water bath. All samples were separated by SDS-PAGE and analyzed by autoradiography. For in vitro kinase assays using MOB1 as substrate, 0.5 nM MST2 was incubated with 0.4 µM recombinant purified MOB1 in the kinase reaction buffer. The phosphorylated MOB1 was analyzed by quantitative Western blotting using a specific antibody against pT35 of MOB1 (Cell Signaling). The signals were quantified with the Odyssey LI-COR imaging system. For in vitro kinase assays using GST-MST2 D146N as substrate, the kinase reaction was performed with MST2 KD for 30 min at room temperature.

For assaying MST2-SAV1 autophosphorylation, 2.5 µM MST2-SAV1-ΔN198 or MST2-SAV1-ΔN268 was dephosphorylated by 0.5 µM PP2A A-C core complex for 30 min at room temperature. Okadaic acid was then added to a final concentration of 5 µM and the reaction mixture was incubated on ice for another 30 min. The reaction mixture was diluted by a ratio of 1:5 at room temperature with a kinase reaction buffer containing 50 mM Tris (pH 7.5), 150 mM NaCl, 5 mM MgCl$_2$, 2 mM ATP, and 1 mM DTT. At different time intervals, autophosphorylation was terminated by adding 2X SDS loading buffer and boiling for 5 min in 100 °C water bath. All samples were separated by SDS-PAGE and blotted by a specific antibody against pT180 of MST2.

## In vitro PP2A inhibition assay

For PP2A A-C phosphatase inhibition assays, 0.1 µM PP2A A-C was mixed with the indicated concentrations of SAV1$^{1-90}$ and incubated at room temperature for 15 min. MST2 kinase domain (KD) was then added to the final concentration of 1 µM in a buffer containing 20 mM Tris (pH 8.0) and 100 mM NaCl. After 30 min incubation, 15 µl reaction mixture was quenched by the addition of the same volume of 2X SDS sample buffer and boiled for 5 min. The remaining pT180 of MST2 was analyzed by quantitative Western blot using a specific antibody against pT180 of MST2.

## Mammalian cell culture, transfection, and RNA interference

293FT cells (Thermo Scientific, catalogue #R70007; not independently authenticated) were cultured in DMEM supplemented with 10% fetal bovine serum, 2 mM L-glutamine and 1% penicillin/streptomycin. MCF10A cells (ATCC, catalogue #CRL-10317; not independently authenticated) were cultured in DMEM/F12 containing 5% horse serum, 20 ng/ml EGF, 0.5 µg/ml hydrocortisone, 100 ng/ml cholera toxin, 10 µg/ml insulin and 1% penicillin/streptomycin. All cells were incubated at 37°C in a humidified 5% CO$_2$ atmosphere. All cell lines were checked by 4',6-diamidino-2-phenylindole (DAPI) staining to ascertain that they were free of mycoplasma contamination. Transient transfection of 293FT cells was performed with Lipojet (Signagen). Cells were collected for further experiments after 24 hr. siRNA transfection was performed with Lipofectamine RNAiMAX (Invitrogen, Waltham, MA) according to manufacturer's instructions. The following siRNAs were synthesized by Dharmacon and used in this study: si*SLMAP* #1, GAAAGCAGCGUCUGAAUAUdTdT; si*SLMAP* #2, GAUCGAAGCCCAGGAGCUAdTdT; si*STRIP1*, GCAGCAAAUUUAUAGGUUAdTdT.

## Generation of knockout cell lines

Gene-specific single-guide RNAs (sgRNAs) were designed using the design tool at https://bench-ling.com. The sgRNAs (SLMAP, CTGTCACGTCTACTCCAAAC; SAV1, GGAGGTGGTTGATCATACCG) were cloned into plentiCRISPR v2 (Addgene). The plentiCRISPR v2-sgRNA, pMD2.G and psPax2 plasmids were co-transfected into 293FT cells with the Lipofectamine 2000 reagent (Invitrogen). Two or three days after transfection, the lentiviral supernatants were harvested and concentrated with Lenti X-concentrator (Clontech). MCF10A cells were infected with the lentiviruses and 4 µg/ml polybrene. Two days after infection, cells were selected with 1 µg/ml puromycin. After two days of selection, single cells were sorted into individual wells of 96-well plates. Single clones were tested by immunoblotting and DNA sequencing.

## Antibodies, immunoblotting, and immunoprecipitation

Rabbit polyclonal MST2 phospho-T336 antibody was raised against the MST2 phospho-peptide with the sequence of ESVGpTMRATC at an on-campus facility. Generation of the MST2 phospho-T378 antibody was described previously (*Ni et al., 2015*). The following antibodies were purchased from the indicated sources: anti-pan Cadherin (C1821, Sigma); anti-STRIP1 (A304-644A) and anti-SLMAP (A304-505A, Bethyl Laboratories Inc.); anti-Tubulin (ab4074), anti-SIKE1 (ab121860) and anti-SLMAP (ab56328, Abcam); anti-MYC (Roche); anti-FLAG (F1804, Sigma); anti-HA (sc-805), anti-PP2A A (sc-6112) and anti-YAP (sc-101199, Santa Cruz Biotechnology); anti-pMST1/2 (T183/T180; 3681), anti-MST1 (3682), anti-GAPDH (2118), anti-MOB1 (13730), anti-pMOB1 (T35;8699), anti-LATS1 (3477), anti-pLATS1/2 (HM; 8654), anti-pYAP (4911), anti-NF2 (12888), anti-PP2A C (2259), anti-rabbit immunoglobulin G (IgG) (H + L) (Dylight 800 or 680 conjugates), anti-mouse IgG (H + L) (Dylight 800 or 680 conjugates) and anti-SAV1 (13301, Cell Signaling). For immunoblotting, cell lysates and immunoprecipitates were analyzed by standard immunoblotting. The membranes were scanned and band intensities were quantified by the Odyssey Infrared Imaging System (LI-COR). We re-blotted the rabbit polyclonal antibody-attached membranes with different mouse monoclonal antibodies, including total YAP, SAV1 and SLMAP antibodies. We also included the GAPDH loading control for each gel, but only showed one representative GAPDH blot for each experiment. We quantified phospho- and total protein bands in a set of three independent experiments. Each band intensity was normalized to the intensity of GAPDH from the same gel. The ratios of normalized phospho-proteins versus normalized total proteins were calculated and plotted. For immunoprecipitation, cells were harvested and lysed with the lysis buffer (20 mM Tris-HCl, pH 7.5, 150 mM NaCl, 0.2% Triton X-100) supplemented with protease inhibitors (Roche) and PhosSTOP (Roche) on ice for 30 min. After incubation, cell lysates were separated by centrifugation. Cleared cell lysates were incubated with anti-FLAG M2 resin (Sigma) for 2 hr at 4°C. After incubation, resins were washed by the washing buffer (20 mM Tris-HCl, pH 7.5, 150 mM NaCl, 0.1 mM EDTA, and 1% Triton X-100) and eluted by SDS sampling buffer. Proteins bound to resins were dissolved in SDS sample buffer, separated by SDS-PAGE, and blotted with the appropriate antibodies.

## Immunofluorescence and subcellular fractionation

Cells on a Lab-Tek II chamber slide were fixed with 4% paraformaldehyde for 20 min, permeabilized with PBS containing 0.2% Triton X-100 (PBS-T) for 20 min, and incubated with PBS-T containing 3% BSA for 30 min. Cells were incubated with the primary antibody in PBS-T containing 3% BSA for 1 hr. Cells were then washed with PBS-T and incubated with fluorescent secondary antibodies. Cells were washed with PBS-T again and mounted in ProLong Gold Antifade reagent with DAPI (Invitrogen). Cells were visualized with a DeltaVision microscope system (Applied Precision). Subcellular fractionation was performed using Mem-PER plus membrane protein extraction kit (Thermo Scientific) according to the manufacturer's instructions.

## RNA isolation and real-time RT-PCR

Total RNA was isolated from cells using the Trizol reagent (Invitrogen). cDNA was obtained by reverse transcription reactions using the Reverse transcription kit (Applied Biosystems). Real-time PCR was performed using qPCR Super Mix-UDG (Invitrogen) and the 7900HT Fast Real-Time PCR System (Applied Biosystems). Relative abundance of mRNA was normalized to GAPDH.

### Statistical analysis

Values are presented as mean ± SEM from at least three biological replicates. Results were evaluated by two-tailed unpaired t tests. The graphs and statistical calculations were performed using Prism (GraphPad).

### Accession codes

Structures and crystallographic data have been deposited at the wwPDB: 6AR0 (apo-SLMAP FHA), 6AR2 (the SLMAP FHA–pMST2 complex), and 6AO5 (the MST2–SAV1 complex).

## Acknowledgements

We thank Hongtao Yu for critical reading of the manuscript and helpful discussion. We are grateful to Yonggang Zheng and Duojia Pan for their suggestions and comments on the paper. We thank Dr. Shih-Chia Tso for assistance with the ITC experiments. Use of the Argonne National Laboratory Structural Biology Center beamlines at the Advanced Photon Source was supported by the US DOE under contract DE-AC02-06CH11357. This work was supported in part by grants from the National Institutes of Health (GM107415 to XL), and the Welch Foundation (I-1932 to XL).

## Additional information

### Funding

| Funder | Grant reference number | Author |
|---|---|---|
| National Institute of General Medical Sciences | GM107415 | Xuelian Luo |
| Welch Foundation | I-1932 | Xuelian Luo |

The funders had no role in study design, data collection and interpretation, or the decision to submit the work for publication.

### Author contributions

Sung Jun Bae, Xuelian Luo, Conceptualization, Data curation, Formal analysis, Supervision, Funding acquisition, Validation, Investigation, Visualization, Methodology, Writing—original draft, Project administration, Writing—review and editing; Lisheng Ni, Data curation, Formal analysis, Validation, Investigation, Visualization, Methodology, Writing—original draft; Adam Osinski, Data curation, Formal analysis, Validation, Investigation, Visualization, Methodology; Diana R Tomchick, Chad A Brautigam, Data curation, Formal analysis, Validation, Visualization, Methodology

### Author ORCIDs

Diana R Tomchick http://orcid.org/0000-0002-7529-4643
Xuelian Luo http://orcid.org/0000-0002-5058-4695

### Decision letter and Author response

Decision letter https://doi.org/10.7554/eLife.30278.021
Author response https://doi.org/10.7554/eLife.30278.022

## Additional files

### Supplementary files

• Transparent reporting form
DOI: https://doi.org/10.7554/eLife.30278.020

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
