## [Decision Letter]

Thank you for submitting your article "SAV1 promotes Hippo kinase activation through antagonizing the PP2A phosphatase STRIPAK" for consideration by *eLife*. Your article has been favorably evaluated by Philip Cole (Senior Editor) and three reviewers, one of whom is a member of our Board of Reviewing Editors. The reviewers have opted to remain anonymous.

The reviewers have discussed the reviews with one another and the Reviewing Editor has drafted this decision to help you prepare a revised submission.

The three reviewers agree that this work represents a potentially significant advance in understanding the role of SAV1 in Hippo signaling. The key finding is that SAV1 helps to activate the Mst1 kinase for phosphorylation of Lats by inhibiting the STRIPAK/SLMAP phosphatase that is also bound to Mst1 in the "off" state, where it prevents phosphorylation of the MST1 activation loop at T180. A variety of potential mechanisms for how it promotes Hippo signaling have been proposed, so the manuscript needs to not only confirm that SAV can inhibit dephosphorylation of MST by STRIPAK, but also compare the proposed mechanism and discuss its relative importance compared to previously suggested mechanisms by which SAV could influence Hippo signaling. Specifically, several concerns need to be addressed in a revised manuscript:

Essential revisions:

1) There are concerns about the reliability of the blots in Figure panels 2B, 3A, and 7B, and the quantification based on them (2C, 3B). It appears from the size, shape, and orientation of the bands shown that the bands shown come from multiple different gels. In order for the GAPDH loading control to be valid, it needs to come from the same gel. Moreover, there are comparisons of phospho- versus total antibody stains for Mst, Mob, Lats, and Yap, and these too look like they come from different gels – indeed, it seems unlikely that they used the same gels for so many different antibodies. The legends and Materials and methods don't provide a detailed explanation for how the blots were generated and how the quantification was done, but it looks like these experiments were not done properly. The authors need to show the proper controls for each blot and provide a much better explanation of the protocols.

2) The ability of the SLMAP deletion to bypass SAV1 necessity supports their model, but considering other available data the result is somewhat puzzling. The authors report that the N-terminus of Sav, which is known to interact with NF2/Merlin, is required for activation of MST by SAV, but they nonetheless appear to invoke a weak interaction of this region with PP2A as being responsible for the influence SAV on Hippo phosphorylation, which seems to ignore the potential roles of both NF2 and SLMAP. The authors do show an inhibition of PP2A activity on MST with the SAV N terminus, but the effect requires very high levels of peptide so may not be biologically relevant. Scaffolding activity provided by the SLMAP-MST1-SAV1 complex could explain the high concentrations needed in these experiments, but this would need to be discussed. Moreover, in the Discussion they seem to hedge against this mechanism, suggesting instead an indirect mechanism through FERM-containing proteins. If the interaction with NF2 is critical for effects of SAV on MST phosphorylation, it could suggest that any interaction with or recruitment of STRIPAK is not the key mechanism by which SAV regulates MST activity. In any case, a more coherent and complete discussion of these points is needed.

3) The interpretation of the structural data shown in Figure 4 characterizing interaction between SAV and MST is incomplete. Data supporting competition between Rassf proteins and SAV for binding to Hpo/Mst was described several years ago by the Tapon lab. The Rassf interaction could provide an alternative mechanism for promoting recruitment of STRIPAK, as proposed by the Tapon lab, but this isn't discussed by the authors, and Rassf isn't depicted in their model Figure 7 A more thorough and balanced discussion is needed here.

4) The experiment examining activation of Mst1 in the cytoplasm raised several concerns. The claim that there is equal activation of membrane and cytosol fractions is based on overexpression of components. Under these conditions they could be getting misleading activation, and perhaps they have saturated all membrane sites. Moreover, the control for membrane fraction is an integral membrane protein rather than a peripheral protein, and it isn't clear that the fractionation protocol will retain such proteins.

---

## [Author Response]

Essential revisions:1) There are concerns about the reliability of the blots in Figure panels 2B, 3A, and 7B, and the quantification based on them (2C, 3B). It appears from the size, shape, and orientation of the bands shown that the bands shown come from multiple different gels. In order for the GAPDH loading control to be valid, it needs to come from the same gel. Moreover, there are comparisons of phospho- versus total antibody stains for Mst, Mob, Lats, and Yap, and these too look like they come from different gels – indeed, it seems unlikely that they used the same gels for so many different antibodies. The legends and Materials and methods don't provide a detailed explanation for how the blots were generated and how the quantification was done, but it looks like these experiments were not done properly. The authors need to show the proper controls for each blot and provide a much better explanation of the protocols.

We thank the reviewers for this insightful comment. In our original experiments, we indeed used multiple different gels because we need to blot with so many different antibodies. Since most phospho-specific antibodies and antibodies against the total protein were rabbit polyclonal antibodies from commercial companies, we could not perform multiplexed blots (with secondary antibodies linked to different fluorophores) on the same membrane. Stripping and re-blotting the same membrane is a potential method that allows one to use a single gel for multiple antibodies in immunoblotting. However, we do not like this method, as we cannot be confident that the membrane-attached antibody can be equally and totally stripped each time. If this were not the case, it could lead to devastating artifacts. Thus, on balance, the use of multiple gels for blotting with different antibodies is the most appropriate and indeed widely accepted practice.

As a general practice in the lab, we routinely include loading controls for each gel. However, as pointed out by the reviewers, we did not use the GAPDH loading control for every gel in this study, because we did not observe significant variations from gel to gel using Ponseau S staining of the membranes. We agree with the reviewer that a proper control loading is needed for every blot to ensure scientific rigor. Therefore, we have re-performed the experiments in Figure 2, Figure 3, and 7B, each with multiple biological repeats. Whenever possible, we re-blotted the same membrane first with rabbit polyclonal phospho-specific antibodies and then with mouse monoclonal antibodies against the relevant total proteins (e.g. YAP, SAV1 and SLMAP). Secondary antibodies against rabbit and mouse IgGs coupled to different fluorophores were used to detect the signals using the Odyssey Infrared Imaging System (LI-COR). We also included the GAPDH loading control for each gel, but only one representative GAPDH blot was shown for each experiment. These new results are presented in the revised Figure 2, Figure 3, and 7B. Furthermore, we quantified phospho- and total protein bands in a set of three independent experiments. Each band intensity was normalized against the intensity of GAPDH from the same gel. The ratios of normalized phospho-proteins versus normalized total proteins were calculated and shown. These new results were presented in the revised Figure 2 and Figure 3. We have also modified the Materials and methods accordingly. The results support our original conclusions.

2) The ability of the SLMAP deletion to bypass SAV1 necessity supports their model, but considering other available data the result is somewhat puzzling. The authors report that the N-terminus of Sav, which is known to interact with NF2/Merlin, is required for activation of MST by SAV, but they nonetheless appear to invoke a weak interaction of this region with PP2A as being responsible for the influence SAV on Hippo phosphorylation, which seems to ignore the potential roles of both NF2 and SLMAP. The authors do show an inhibition of PP2A activity on MST with the SAV N terminus, but the effect requires very high levels of peptide so may not be biologically relevant. Scaffolding activity provided by the SLMAP-MST1-SAV1 complex could explain the high concentrations needed in these experiments, but this would need to be discussed. Moreover, in the Discussion they seem to hedge against this mechanism, suggesting instead an indirect mechanism through FERM-containing proteins. If the interaction with NF2 is critical for effects of SAV on MST phosphorylation, it could suggest that any interaction with or recruitment of STRIPAK is not the key mechanism by which SAV regulates MST activity. In any case, a more coherent and complete discussion of these points is needed.

We thank the reviewers for this very insightful comment. We have revised our Discussion to emphasize the scaffolding model. We also briefly discuss how the SAV1-STRIPAK^SLMAP^ antagonism can be potentially regulated by NF2 or EX during Hippo activation.

3) The interpretation of the structural data shown in Figure 4 characterizing interaction between SAV and MST is incomplete. Data supporting competition between Rassf proteins and SAV for binding to Hpo/Mst was described several years ago by the Tapon lab. The Rassf interaction could provide an alternative mechanism for promoting recruitment of STRIPAK, as proposed by the Tapon lab, but this isn't discussed by the authors, and Rassf isn't depicted in their model Figure 7 A more thorough and balanced discussion is needed here.

We thank the reviewers for this insightful comment. We have revised the main text and added the relevant references about the competition between RASSF and SAV for Hippo /MST binding accordingly. We also added more discussion on the alternative mechanism by which RASSF can recruit STRIPAK to Hippo/MST for its inactivation. Because our model in Figure 7 highlights SAV1 regulation on MST activity shown by our own data, we did not add RASSF in our model. Although our results are entirely consistent with a role of RASSF in STRIPAK recruitment, we do not have direct data on RASSF to support this conclusion. Adding RASSF in the model might mislead the readers into thinking that we had examined RASSF in this study. Moreover, as it stands, the model is already complicated. We hope that the reviewers would understand.

4) The experiment examining activation of Mst1 in the cytoplasm raised several concerns. The claim that there is equal activation of membrane and cytosol fractions is based on overexpression of components. Under these conditions they could be getting misleading activation, and perhaps they have saturated all membrane sites. Moreover, the control for membrane fraction is an integral membrane protein rather than a peripheral protein, and it isn't clear that the fractionation protocol will retain such proteins.

We thank the reviewers for this helpful comment. In our original experiments, we showed that SAV1 specifically activated overexpressed MST2. For example, many SAV1 mutants, especially WW domain deletion mutants (which are not related to MST2 binding), could not activate overexpressed MST2. Meanwhile, co-expression of RASSF1A could not enhance MST2 activation. We performed the membrane fractionation assay because we wanted to check where MST2 was activated by SAV1 inside the cells and if SAV1 affected MST2 localization in our experimental conditions. The result showed that MST2 was localized to both cytosol and membranes, and MST2 was equally activated by co-expression of SAV1 at both locations. As suggested by the reviewers, we performed the cytosol-membrane fractionation assay again using three different cellular localization markers: Tubulin for cytosolic proteins, Pan Cadherin for integral membrane proteins, and NF2 for peripheral membrane proteins. The results showed that the membrane fraction used in this assay contained both peripheral and integral membrane proteins. These new data were presented in the revised Figure 5—figure supplement 2. The relevant legend and method were revised accordingly.